# Serine hydroxymethyltransferase as a potential target of antibacterial agents acting synergistically with one-carbon metabolism-related inhibitors

Yuko Makino[1], Chihiro Oe[1], Kazuya Iwama[1], Satoshi Suzuki[1], Akie Nishiyama[1], Kazuya Hasegawa[2], Haruka Okuda[3], Kazushige Hirata[1,4], Mariko Ueno[1], Kumi Kawaji[3], Mina Sasano[3], Emiko Usui[3], Toshiaki Hosaka [5], Yukako Yabuki[5], Mikako Shirouzu [5], Makoto Katsumi[4], Kazutaka Murayama[5,6], Hironori Hayashi [3,7✉] & Eiichi N. Kodama[1,3,7,8]

Serine hydroxymethyltransferase (SHMT) produces 5,10-methylenetetrahydrofolate ($CH_2$-THF) from tetrahydrofolate with serine to glycine conversion. SHMT is a potential drug target in parasites, viruses and cancer. (+)-SHIN-1 was developed as a human SHMT inhibitor for cancer therapy. However, the potential of SHMT as an antibacterial target is unknown. Here, we show that (+)-SHIN-1 bacteriostatically inhibits the growth of *Enterococcus faecium* at a 50% effective concentration of $10^{-11}$ M and synergistically enhances the antibacterial activities of several nucleoside analogues. Our results, including crystal structure analysis, indicate that (+)-SHIN-1 binds tightly to *E. faecium* SHMT (*efm*SHMT). Two variable loops in SHMT are crucial for inhibitor binding, and serine binding to *efm*SHMT enhances the affinity of (+)-SHIN-1 by stabilising the loop structure of *efm*SHMT. The findings highlight the potency of SHMT as an antibacterial target and the possibility of developing SHMT inhibitors for treating bacterial, viral and parasitic infections and cancer.

[1] Department of Infectious Diseases, Tohoku University Graduate School of Medicine, 2-1, Seiryo-machi, Aoba-ku, Sendai, Miyagi 980-8575, Japan. [2] Structural Biology Division, Japan Synchrotron Radiation Research Institute, 1-1, Sayo-chou, Hyogo, Japan. [3] Division of Infectious Diseases, International Research Institute of Disaster Science, Tohoku University, 2-1, Seiryo-machi, Aoba-ku, Sendai, Miyagi 980-8575, Japan. [4] Division of Clinical Laboratory, Department of Clinical Laboratory Medicine, Tohoku University Hospital, 1-1, Seiryo-machi, Aoba-ku, Sendai, Miyagi 980-8574, Japan. [5] Laboratory for Protein Functional and Structural Biology, RIKEN Center for Biosystems Dynamics Research, Suehiro 1-7-22, Tsurumi, Yokohama, Japan. [6] Division of Biomedical Measurements and Diagnostics, Graduate School of Biomedical Engineering, Tohoku University, Sendai, Japan. [7] Department of Intelligent Network for Infection Control, Tohoku University Graduate School of Medicine, 2-1, Seiryo-machi, Aoba-ku, Sendai, Miyagi 980-8575, Japan. [8] Tohoku Medical Megabank Organization, Tohoku University, 2-1, Seiryo-machi, Aoba-ku, Sendai, Miyagi 980-8575, Japan. ✉email: hhayashi@med.tohoku.ac.jp

One-carbon (1C) metabolism is an essential pathway for constructing fundamental biomolecules necessary for prokaryotic and eukaryotic cell growth and proliferation[1–4]. 1C metabolism encompasses a complex metabolic network involving folate-dependent chemical reactions. In 1C metabolism, folate compounds are cyclically used or recycled by folate-dependent enzymes, such as dihydrofolate reductase (DHFR), serine hydroxymethyltransferase (SHMT) and thymidylate synthase (TS). DHFR converts dihydrofolate to tetrahydrofolate (THF). SHMT produces 5,10-methylenetetrahydrofolate ($CH_2$-THF) from THF by converting serine into glycine. $CH_2$-THF acts as a 1C unit carrier and is used as a fundamental building block for essential biomolecules (e.g., amino acids and nucleotides)[2,4]. Glycine is used for purine synthesis in several pathways downstream of 1C metabolism. In addition, TS converts deoxyuridine monophosphate to thymidine monophosphate (dTMP) by transferring the 1C unit from $CH_2$-THF. Subsequently, dTMP is phosphorylated to form thymidine triphosphate, a DNA nucleotide.

Suppressing 1C metabolism using chemicals is effective in cancer and antibacterial therapies. Pemetrexed (PMX) and methotrexate (MTX) are folate antagonists in human cells and inhibit DHFR in cancer cells[5], whereas 5-fluorouracil (5-FU) targets nucleotide metabolism, including thymidine synthesis[6]. These compounds are major clinical anticancer agents that have been used as frontline chemotherapies to treat various cancers. Trimethoprim (TMP) is used as a bacterial DHFR inhibitor in antibacterial therapy. We reported previously that 5-fluoro-2′-deoxyuridine (5-FdU) inhibits bacterial TS and induces thymineless death in *Staphylococcus aureus*[6].

SHMT, a pyridoxal 5′-phosphate (PLP)-dependent enzyme, catalyses the retro-aldol cleavage of serine to glycine and converts THF into $CH_2$-THF[7,8]. The glycine and $CH_2$-THF supplied by SHMT are essential for pyrimidine, purine and amino acid syntheses[9,10]. Therefore, drugs targeting SHMT can block multiple metabolic pathways downstream of 1C metabolism (e.g., DNA and RNA synthesis), indicating that SHMT is a potential target of therapeutic agents directed at various viruses, bacteria and parasite infections, as well as cancer. Indeed, severe acute respiratory syndrome coronavirus 2 (SARS-CoV-2) has been shown to induce purine synthesis to support the synthesis of viral subgenomic RNA by hijacking serine metabolism through human SHMT1[11]. Moreover, SHMT inhibitors under development are promising therapeutic agents against *Plasmodium* parasites[12,13]. Furthermore, novel chemical tools for cancer chemotherapy have been developed that inhibit SHMT activity by targeting the production of 1C units from serine, a primary 1C unit source in cancer cells[14]. The pyrazolopyran derivatives (+)-SHIN-1 and SHMT-IN-2 have anticancer activities against large, diffuse B-cell lymphoma by inhibiting human SHMT1/2 activity[14]. Sertraline (SER), a neuromodulatory drug, has also been found to suppress breast tumour growth by inhibiting SHMT1/2[15,16].

In contrast to cancer treatment, the effect of SHMT inhibition on prokaryotic cells is unclear. The *Enterococcus* genus requires folate-related compounds for growth and incorporates these compounds[17]. In particular, *Enterococcus faecium* has been used for studying the bacterial folate cycle and for drug screening because this bacterium grows in folate-free media that includes amino acids and nucleic acids (purine and pyrimidine bases)[18]. *E. faecium* is susceptible to bacterial TS-inhibiting pyrimidine analogues such as 5-FdU[19]. Thus, *E. faecium* is a suitable bacterium for determining the potential of SHMT as an antibacterial target and for investigating the synergistic effects of inhibiting 1C metabolism using a combination of DHFR, TS and SHMT inhibitors.

Here, we investigated SHMT as an antibacterial target using *E. faecium* and (+)-SHIN-1. (+)-SHIN-1 activity was highly potent against *E. faecium* with 50% effective concentration ($EC_{50}$) values on the order of $10^{-11}$ M. We found that excess deoxythymidine (dT) reduced (+)-SHIN-1 activity, suggesting that the major antibacterial mechanism involved inhibiting physiological thymidine synthesis by blocking the supply of the 1C-unit carrier of 1C metabolism, $CH_2$-THF. Differential scanning fluorimetry and crystal structural analyses revealed that (+)-SHIN-1 acts as an *E. faecium* SHMT (*efm*SHMT) inhibitor and that the biphenyl and hydroxy groups in (+)-SHIN-1 are important for high-affinity binding to *efm*SHMT. *E. faecium* is also susceptible to DHFR inhibitors, such as PMX, MTX and raltitrexed (RTX), and 5-FdU, which is also reported as a TS inhibitor[19]. Thus, *E. faecium* is susceptible to compounds inhibiting three major enzyme types in 1C metabolism. We found that (+)-SHIN-1 enhances the antibacterial activity of nucleoside analogues, suggesting that it has a synergistic effect when used with DNA and RNA synthesis inhibitors. These results shed light on the potential of SHMT as an antibacterial target and support the development of SHMT inhibitors as therapeutic agents for bacterial, viral and parasite infections and cancer.

## Results

**Bacterial sensitivity to 1C metabolism inhibitors**. The potency of SHMT as an antibacterial target was evaluated by determining the antibacterial activities of several 1C metabolism inhibitors on *E. faecium*, *E. faecalis*, *S. aureus* and *E. coli* (Fig. 1a, Table 1).

**Fig. 1 Chemical structures of 1C metabolism-related compounds.** Chemical structures are shown. SHMT-IN-2; SHMT inhibitor 2, PMX pemetrexed, MTX methotrexate, RTX raltitrexed, TMP trimethoprim, 5-FdU 5-fluoro-2′-deoxyuridine, 5-FUrd 5-fluorouridine, and SER sertraline.

**Table 1 Antibacterial activities of folate-mediated 1C metabolism inhibitors against four microorganisms.**

| Compounds | $EC_{50}$ ($\mu$M) (p$EC_{50}$ ± S.E.) | | | |
|---|---|---|---|---|
| | E. faecium | E. faecalis | S. aureus | E. coli |
| VCM ($\mu$g/mL) | 0.31 (3.5 ± 0.0) | 3.2 (2.5 ± 0.0) | 0.36 (3.4 ± 0.1) | >100 (<1.0) |
| (+)-SHIN-1 | 0.000031 (10.5 ± 0.1) | >2 (<5.7) | >2 (<5.7) | >2 (<5.7) |
| SHIN-1 | 0.00050 (9.3 ± 0.1) | >2 (<5.7) | >2 (<5.7) | >2 (<5.7) |
| SHMT-IN-2 | 2.9 (5.5 ± 0.0) | >100 (<4.0) | ≥100ᵃ (≤4.0) | >100 (<4.0) |
| SER | 17 (4.8 ± 0.0) | 29 (4.5 ± 0.0) | 29 (4.5 ± 0.0) | 29 (4.5 ± 0.0) |
| PMX | 0.000052 (10.3 ± 0.1) | >100 (<4.0) | >100 (<4.0) | >100 (<4.0) |
| MTX | 0.0036 (8.4 ± 0.0) | 14 (4.8 ± 0.0) | >100 (<4.0) | >100 (<4.0) |
| RTX | 0.013 (7.9 ± 0.1) | >100 (<4.0) | >100 (<4.0) | >100 (<4.0) |
| TMP | 0.0028 (8.6 ± 0.2) | 0.27 (6.6 ± 0.0) | 0.29 (6.5 ± 0.0) | 0.17 (6.8 ± 0.0) |
| 5-FdU | 0.029 (7.5 ± 0.1) | 0.31 (6.5 ± 0.0) | 0.0089 (8.1 ± 0.2) | 10 (5.0 ± 0.0) |
| 5-FUrd | 0.073 (7.1 ± 0.1) | 0.38 (6.4 ± 0.1) | 0.030 (7.5 ± 0.0) | 10 (5.0 ± 0.0) |

The 50% effective concentration ($EC_{50}$) represents the range determined from at least three independent experiments. p$EC_{50}$ is log-transformed $EC_{50}$.
*VCM* vancomycin, *SER* sertraline, *PMX* pemetrexed, *MTX* methotrexate, *RTX* raltitrexed, *TMP* trimethoprim, *5-FdU* 5-fluoro-2′-deoxyuridine, *5-FUrd* 5-fluorouridine.
ᵃOne of three $EC_{50}$ values were over 100 $\mu$M and the others were 40–60 $\mu$M. All data represent the mean ± standard deviation (n = 3 and n = 5 for (+)-SHIN-1 and PMX). The E. faecium, E. faecalis, S. aureus and E. coli strains are JCM5804, JCM7783, ATCC29213 and ATCC25922, respectively.

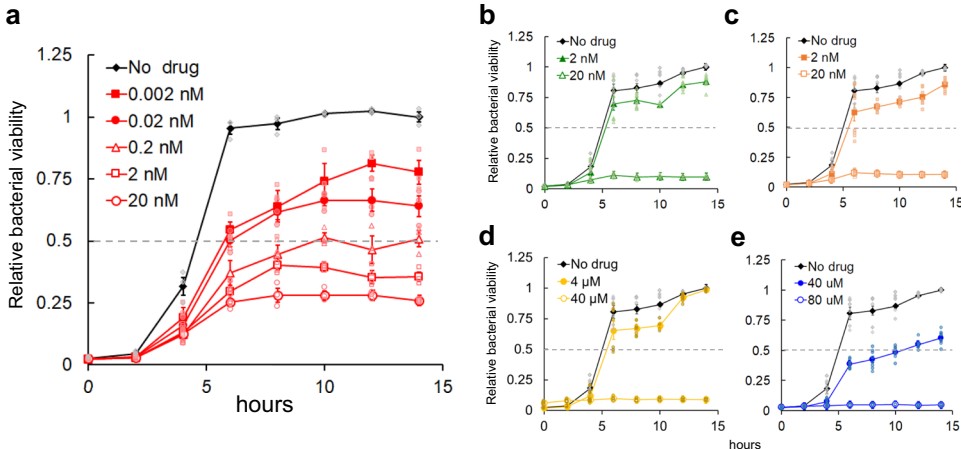

**Fig. 2 The results of the bacterial viability assay. a** Bacterial viability in the absence and presence of different (+)-SHIN-1 concentrations. Grey and light red-coloured symbols are the values for calculating the average values and standard error of each point. (n = 3) **b–e** Bacterial viabilities in the absence and presence of different MTX, TMP, SER and SHMT-IN-2 concentrations, respectively. Light green, light orange, dark yellow and light blue symbols show the values for calculating the average values and standard error of each point (n = 8, n = 4 at 10 h) Error bars mean standard error.

SHIN-1 and (+)-SHIN-1 displayed very strong antibacterial activities with $EC_{50}$ values of $10^{-10}$ to $10^{-11}$ M against *E. faecium*, whereas SHMT-IN-2 had weaker activity ($EC_{50}$: 2.9 × $10^{-6}$ M). *E. faecium* was susceptible to human folate metabolism inhibitors (PMX, MTX and RTX). PMX and MTX displayed especially potent activities ($EC_{50}$ of $10^{-9}$–$10^{-10}$ M). However, they were inactive against *E. faecalis*, *S. aureus* and *E. coli*, even at 100 $\mu$M. Furthermore, the Gram-positive bacteria (*E. faecium*, *E. faecalis* and *S. aureus*) were also susceptible to 5-FdU, which acts as a bacterial TS inhibitor, as we have reported[19]. SER inhibited the growth of all bacteria using this assay with $EC_{50}$ values of $10^{-5}$ M.

**Determination of time-dependent bacterial viability with or without inhibitors.** We evaluated the time-dependent change in bacterial viability in the presence and absence of each inhibitor to quantitate the bacterial cell growth carefully and determine the inhibition mechanism of the compounds (Fig. 2)[6]. The results revealed that (+)-SHIN-1, MTX, TMP and SER inhibited the growth of *E. faecium* with similar $EC_{50}$ values to those of turbidity detection ($EC_{50}$ = 0.00023, 0.0084, 0.0088 and 13 $\mu$M, respectively). SHMT-IN-2 showed weaker activity than the turbidity

detection result. Notably, time-dependent bacterial viabilities indicated that (+)-SHIN-1 acts bacteriostatically rather than function in a bactericidal manner against *E. faecium*.

**Cell cytotoxicity of the 1C metabolism inhibitors.** Antibacterial activity was compared with the $CC_{50}$ of each compound by employing several human cell lines (Caski, Hep-2 and Calu-3) and calculating the selectivity index (S.I.) values, which we defined as the ratio of a compound's $CC_{50}$ against each cell line and the $EC_{50}$ against *E. faecium* (Supplementary Table S2). As a result, the SI value for (+)-SHIN-1 with each cell line exceeded 45,000, indicating high bacterial selectivity. TMP also showed high bacterial selectivity. The SI value for PMX differed in each cell line, but the values exceeded 14,000. However, MTX and SHMT-IN-2 were less selective than (+)-SHIN-1, TMP and PMX. SER displayed no selectivity.

**Antibacterial assay with excess concentrations of serine, glycine and physiological nucleosides.** The antibacterial mechanism of (+)-SHIN-1 and other compounds was determined by performing antibacterial assays with excess concentrations of folic

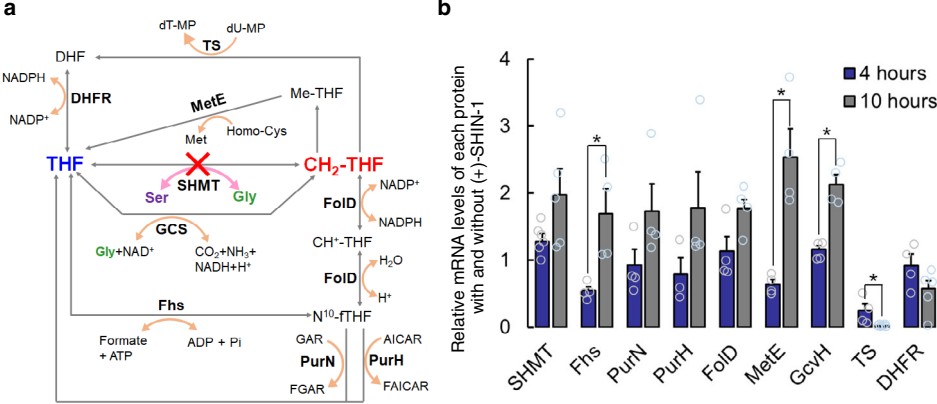

**Fig. 3 The postulated folate cycle in *E. faecium* and changes in mRNA levels of folate-binding proteins upon addition of (+)-SHIN-1. a** *E. faecium* genome [Genbank code: UFYJ01000001.1] codes for several proteins related to the folate cycle. GCS and Fhs may support production of THF and CH$_2$-THF under SHMT-inhibition conditions. **b** The relative mRNA expression levels of each folate-binding protein with and without (+)-SHIN-1. Four hours after adding 2 μM (+)-SHIN-1, mRNAs for expression of SHMT and GcvH increased and those of Fhs, MetE and TS decreased. In contrast, 10 h after adding 2 μM (+)-SHIN-1, mRNA expression levels of folate-binding proteins, except TS and DHFR, tended to increase. Open circles are the values for calculating the average values and standard error ($n = 4$). For SHMT at 4 and 10 h, the $n$ number is six ($n = 6$) and five ($n = 5$) respectively. For PurH at 4 h, the n number is three ($n = 3$). *Asterisk means p-value < 0.05 (P = 0.01 for Fhs and GcvH, and P = 0.04 for MetE and TS) For DHFR at 10 h, the $n$ number is five. TS thymidylate synthetase, DHFR dihydrofolate reductase, GCS glycine cleavage system, MetE cobalamin-independent methionine synthase, Fhs formyltetrahydrofolate synthetase, FolD 5,10-methylene tetrahydrofolate dehydrogenase, PurN glycinamide-RNase-transformylase N, GAR glycinamide ribonucleotide, FGAR formyl glycinamide ribonucleotide, PurH AICAR transformylase, AICAR 5-amino-4-imidazole carboxamide ribotide, FAICAR 5-formamidoimidazole-4-carboxamide ribonucleotide, GcvH glycine cleavage system H protein.

acid, Ser, Gly, inosine and dT when compared with the physiological concentrations of these molecules (Fig. 3a). In the presence of 100 mM Ser, the antibacterial activities of (+)-SHIN-1 and SHIN-1 were 10- and 5-fold weaker than when Ser was absent (Table 2). The bacterial viability assay showed that *E. faecium* growth was affected in the presence of 100 mM Ser (Supplementary Fig. S1) and that the antibacterial activity of (+)-SHIN-1 with 100 mM Ser was weaker when compared with the antibacterial activity of (+)-SHIN-1 in the absence of Ser. The decrease in antibacterial activity with Ser was not caused by competition between (+)-SHIN-1 and Ser. PMX also displayed weaker antibacterial activity in the presence of 100 mM Ser. The other compounds, including SHMT-IN-2 and SER, showed similar activities with/without 100 mM Ser. In the presence of 10 mM Gly, (+)-SHIN-1 and SHIN-1 displayed 3- and 2-fold weaker activities than when Gly was absent. The activity of 5-fluorouridine (5-FUrd) was slightly stronger in the presence versus absence of Ser and Gly. In the presence of 50 and 100 mM Gly, the antibacterial activity of each drug was not detectable because Gly caused cell toxicity (Supplementary Fig. S2). In the presence of 1 mM inosine, compounds except (+)-SHIN-1 and 5-FUrd displayed similar activities to those when inosine was absent. As shown in Fig. 3a, an excess amount of Ser, which is an SHMT substrate, can counteract the activity of (+)-SHIN-1. Abundant inosine levels potentially offset the nucleotide imbalance caused by decreased levels of purine bases. The antibacterial activities of 5-FUrd and (+)-SHIN-1 with 1 mM inosine were weaker (4- and 13-fold, respectively) than those without inosine. In the presence of 1 mM dT, tested compounds except SHMT-IN-2 and SER lost their antibacterial activities. This observation suggests that 1 C metabolism inhibitors cause thymine starvation in *E. faecium*, but the antibacterial mechanism of SHMT-IN-2 and SER differs from these compounds[20,21]. The antibacterial activity of (+)-SHIN-1, SHIN-1 and folate analogues decreased when 0.1 mM folate (FA) was present, whereas the antibacterial activity of SHMT-IN-2 and SER was independent of FA (Table 2). Thus, SHMT-IN-2 and SER do not target folate cycle-related metabolic enzymes, including SHMT.

**Thermal stability of *efm*SHMT with or without each compound.** We examined the interaction between *efm*SHMT and each compound using differential scanning fluorimetry (DSF)[22,23]. As illustrated in Fig. 3, the thermal stability of *efm*SHMT only shifted in the presence of (+)-SHIN-1 ($\Delta T_{\mathrm{m}} = 7\,°C$), indicating that (+)-SHIN-1 bound to *efm*SHMT. SHMT-IN-2, SER and MTX did not stabilise *efm*SHMT significantly, whereas PMX ($\Delta T_{\mathrm{m}} = 0.9\,°C$) stabilised *efm*SHMT slightly (Supplementary Fig. S3). These data support the results of the competition assays. Therefore, (+)-SHIN-1 is an *efm*SHMT inhibitor and SHMT-IN-2, SER and MTX are not. PMX binds to *efm*SHMT, but the affinity is probably quite weak. The DSF results using *ec*SHMT and each compound indicated that (+)-SHIN-1 bound to *ec*SHMT (Supplementary Fig. S4), but other compounds did not. Antibacterial assays and DSF results suggested that (+)-SHIN-1 was not captured by *E. coli*. The biphasic nature of the melting curve induced by (+)-SHIN-1 complicated melting point calculations. This biphasic curve may have arisen because of monomer/dimer melting.

**Evaluation of the binding affinities of each compound against *efm*SHMT.** The binding affinities of each compound against *efm*SHMT were evaluated using a binding assay that utilised the 500-nm absorption value derived from quinonoid formation in the SHMT ternary complex of SHMT, Gly and 5-methyltetrahydrofolate (Me-THF) (*efm*SHMT/Gly/Me-THF)[24,25]. In the presence of excess Gly, the *efm*SHMT/Gly/Me-THF complex forms quinonoid, whereas the *efm*SHMT-inhibitor complex does not. Therefore, adding a potential inhibitor to the ternary complex decreases the absorbance at 500 nm (OD$_{500}$) derived from quinonoid if the compound functions as an SHMT inhibitor (Fig. 4a). Using this feature and the Cheng-Prusoff equation, we estimated the $K_{\mathrm{i}}$ of compounds. (+)-SHIN-1 showed a strong binding affinity in the presence of 2.5 or 5 μM *efm*SHMT ($K_{\mathrm{i}} = 0.0088$ and 0.028 μM, respectively) (Fig. 5b). Additionally, PMX bound to *efm*SHMT with weak affinity ($K_{\mathrm{i}} = 117$ μM), whereas SHMT-IN-2, SER and MTX did not bind to *efm*SHMT (Fig. 4b and Supplementary Fig. S5). These results are consistent with the DSF results. However, in this assay, the

**Table 2 Antibacterial activities of E. faecium inhibitors in the presence of excess amounts of serine, glycine, inosine, deoxythymidine or folate.**

| Compounds | 100 mM Serine | | 10 mM Glycine | | 1 mM Inosine | | 1 mM deoxythymidine | | 0.1 M folate | |
|---|---|---|---|---|---|---|---|---|---|---|
| | EC$_{50}$ (µM) (pEC$_{50}$ ± S.E.) | F.I.[a] | EC$_{50}$ (µM) (pEC$_{50}$ ± S.E.) | F.I.[a] | EC$_{50}$ (µM) (pEC$_{50}$ ± S.E.) | F.I.[a] | EC$_{50}$ (µM) (pEC$_{50}$ ± S.E.) | F.I.[a] | EC$_{50}$ (µM) (pEC$_{50}$ ± S.E.) | F.I.[a] |
| VCM (µg/mL) | 0.30 (3.5 ± 0.0) | 1 | 0.29 (3.5 ± 0.0) | 1 | 0.34 (3.5 ± 0.2) | 1 | 0.42 (3.4 ± 0.2) | 1 | 0.31 (3.5 ± 0.0) | 1 |
| (+)-SHIN-1 | 0.00043 (9.4 ± 0.1) | 10 | 0.00012 (9.9 ± 0.0) | 10 | 0.00036 (9.4 ± 0.1) | 3 | >2 (<5.7) | >4 × 10$^4$ | >2 (<5.7) | >4 × 10$^4$ |
| SHIN-1 | 0.0034 (8.5 ± 0.1) | 5 | 0.0012 (8.9 ± 0.0) | 10 | 0.0052 (8.3 ± 0.3) | 2 | >2 (<5.7) | >3 × 10$^3$ | >2 (<5.7) | >3 × 10$^3$ |
| SHMT-IN-2 | 4.0 (5.4 ± 0.0) | 1 | 3.0 (5.5 ± 0.0) | 1 | 3.1 (5.5 ± 0.0) | 1 | 2.9 (5.5 ± 0.0) | 1 | 2.9 (5.5 ± 0.0) | 1 |
| SER | 19 (4.7 ± 0.1) | 1 | 22 (4.6 ± 0.0) | 1 | 20 (4.7 ± 0.1) | 1 | 17 (4.8 ± 0.0) | 1 | 17 (4.8 ± 0.0) | 1 |
| PMX | 0.00025 (9.6 ± 0.0) | 5 | 0.000062 (10.2 ± 0.0) | 5 | 0.000074 (10.1 ± 0.1) | 2 | >100 (<4.0) | >2 × 10$^6$ | 1.8 (5.7 ± 0.0) | 4 × 10$^6$ |
| MTX | 0.0026 (8.6 ± 0.0) | 0.7 | 0.0028 (8.6 ± 0.0) | 0.8 | 0.0024 (8.6 ± 0.0) | 0.8 | >100 (<4.0) | >2 × 10$^4$ | 8.5 (5.1 ± 0.1) | 2 × 10$^3$ |
| RTX | 0.014 (7.9 ± 0.1) | 1 | 0.010 (8.0 ± 0.1) | 0.8 | 0.0082 (8.1 ± 0.2) | 0.6 | >100 (<4.0) | >7 × 10$^3$ | >100 (<4.0) | >7 × 10$^3$ |
| TMP | 0.0033 (8.5 ± 0.0) | 0.8 | 0.0021 (8.7 ± 0.1) | 0.8 | 0.024 (8.6 ± 0.0) | 0.6 | 54 (4.3 ± 0.1) | >2 × 10$^4$ | 0.041 (7.3 ± 0.0) | 14 |
| 5-FdU | 0.032 (7.5 ± 0.1) | 1 | 0.0064 (8.2 ± 0.0) | 0.2 | 0.024 (7.6 ± 0.0) | 0.6 | >100 (<4.0) | >2 × 10$^3$ | 0.19 (6.7 ± 0.1) | 5 |
| 5-FUrd | 0.098 (7.0 ± 0.1) | 1 | 0.065 (7.1 ± 0.0) | 0.8 | 0.72 (6.1 ± 0.0) | 9 | 21 (4.7 ± 0.0) | 259 | 0.27 (6.6 ± 0.0) | 3 |

The 50% effective concentration (EC$_{50}$) represents the range determined from at least three independent experiments. pEC$_{50}$ is log-transformed EC$_{50}$. All data represent the mean ± standard deviation ($n = 3$ and $n = 6$ for (+)-SHIN-1). aFold increase is the ratio of the EC$_{50}$ value with an excess amount of supplement and the EC$_{50}$ value without any supplement against E. faecium, as described in Table 1. The E. faecium strain is JCM5804.
VCM vancomycin, SER sertraline, PMX pemetrexed, MTX methotrexate, RTX raltitrexed, TMP trimethoprim, 5-FdU 5-fluoro-2′-deoxyuridine, 5-FUrd 5-fluorouridine.

detection limit concentration of *efm*SHMT was 2.5 µM, and the concentration was too high to evaluate the affinity of (+)-SHIN-1 accurately. In contrast, the lowest SHMT concentration used represents the lower limit $K_i$ of (+)-SHIN-1 determined in this assay. Therefore, the actual $K_i$ is probably lower than 0.0088 µM, indicating the extremely strong affinity of (+)-SHIN-1 toward *efm*SHMT. The DSF and the binding assay results showed that the main target of (+)-SHIN-1 is SHMT in *E. faecium*, which is the same as in humans. Furthermore, the binding affinity of (+)-SHIN-1 toward *efm*SHMT was enhanced in the presence of Ser (Supplementary Fig. S6).

**Structural comparisons between *efm*SHMT and three reported SHMTs.** We solved the structures of *efm*SHMT in complex with pyridoxal 5′-phosphate (PLP) and (+)-SHIN-1 in the absence or presence of Ser (Fig. 5 and Table 3) at 2.28 and 1.90 Å resolutions, respectively, to characterise SHIN-1 binding to *efm*SHMT. In addition to these structures, the *efm*SHMT/Gly/Me-THF complex structure was determined at 2.62 Å resolution. DLS measurements indicated that the hydrodynamic radius of *efm*SHMT was 4.209 nm with an evaluated molecular weight of 97.2 kDa (theoretical monomer molecular weight is 44.7 kDa), suggesting *efm*SHMT forms a dimer in solution. In the crystal structures, we observed that the asymmetric unit includes two molecules that form a dimer.

In the complex structure with (+)-SHIN-1, electron density was observed in both binding pockets of the protein dimer (Fig. 5b), and the (+)-SHIN-1 binding sites were well resolved. The C4′ carbon in the PLP ring was covalently linked as a Schiff's base to K226, as previously reported for the *hu*SHMT structure[14]. The solved structure showed that (+)-SHIN-1 binds to the folate-binding site of *efm*SHMT. The structure of the binding site of *efm*SHMT was compared with previously reported structures of human, *E. coli* and *Plasmodium vivax* SHMTs, *hu*SHMT2, *ec*SHMT and *pv*SHMT, respectively, by superimposing the *efm*SHMT catalytic site on each SHMT structure (PDB IDs: 5V7I, 1DFO and 5GVN, respectively) (Fig. 6). The backbone root-mean-square deviations of the superimposed structures were 1.36 Å for *hu*SHMT2, 0.97 Å for *ec*SHMT and 1.35 Å for *pv*SHMT. In the catalytic site, we found that two variable loop regions encompassing amino acids (aa) 115–137 (115-loop) and 343–357 (343-loop) in *efm*SHMT differed from each other (Fig. 6). The aa sequences of the 115- and 343-loops in *efm*SHMT differ markedly from those in *hu*SHMT2 and *pv*SHMT, with an aa homology of <40% (Supplementary Table S4). In contrast, the 115-loop in *ec*SHMT has a similar structure to that found in *efm*SHMT. The 343-loop structure of *efm*SHMT differs from those of the other enzymes. The homologies of the 115- and 343-loops between *efm*SHMT and *ec*SHMT were 74% and 60%, respectively (Supplementary Table S3). As previously reported, we found that the 343-loop is flexible, and its position depends on the compounds that bind to the catalytic site[13,26]. The other loops in the catalytic site are inflexible and occupy conserved positions[13]. Therefore, the difference in the 343-loop structure between *efm*SHMT and *ec*SHMT is possibly related to substrate binding to each SHMT, whereas the differences in the 115-loop structures are associated with aa sequence differences.

**Ligand interactions among efmSHMT complexes.** We identified the following interactions between the *efm*SHMT and (+)-SHIN-1 complex. The exocyclic amine in (+)-SHIN-1 forms hydrogen bonds with the main chain of Leu117 and Gly121 (Fig. 5c). This exocyclic amine also interacts with Asn343 and Ser344 through a water molecule. The pyrazole moiety forms a hydrogen bond and van der Waals interactions with the main chain of Leu123 and

**a**

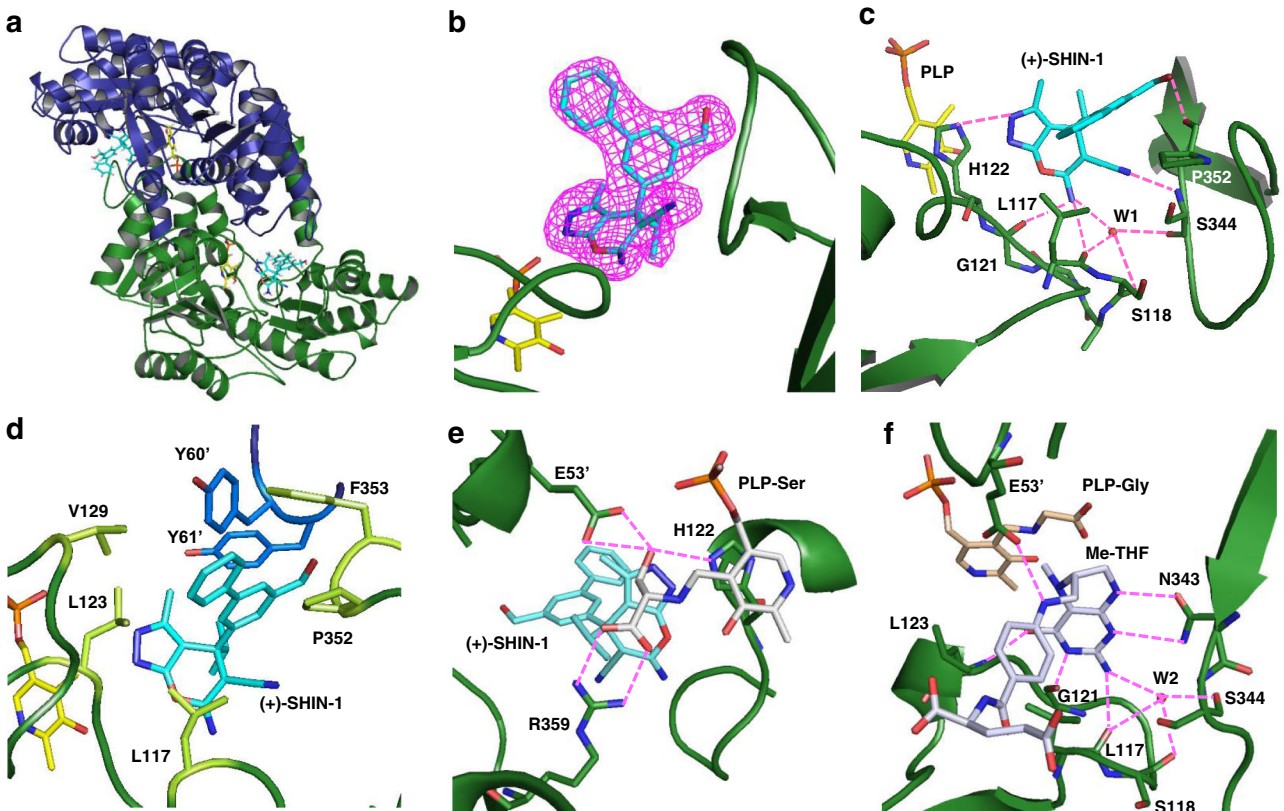

**b**

| Compounds | SHMT | Me-THF | Compounds | |
|---|---|---|---|---|
| | Conc. (µM) (pConc.) | EC$_{50}$ (µM) (pEC$_{50}$ ± pSE ) | Conc. (µM) (pConc.) | $K_i^†$ (µM) (p$K_i$ ± pSE ) |
| No drug | 5 (5.3) | 18 (4.7 ± 0.1) | - | - |
| | 2.5 (5.6) | 9.5 (5.0 ± 0.1) | - | - |
| (+)-SHIN-1 | 5 (5.3) | 4055 (2.4 ± 0.1) | 5 (5.3) | 0.028 (7.6 ± 0.1) |
| | 2.5 (5.6) | 2319 (2.6 ± 0.1) | 2.5 (5.6) | 0.0088 (8.1 ± 0.1) |
| SHMT-IN-2 | 5 (5.3) | 16 (4.8 ± 0.1) | 100 (4.0)* | No inhibition |
| SER | 5 (5.3) | 20 (4.7 ± 0.1) | 400 (3.4) | No inhibition |
| PMX | 5 (5.3) | 49 (4.3 ± 0.1) | 400 (3.4) | 117 (3.9 ± 0.1) |
| MTX | 5 (5.3) | 14 (4.8 ± 0.0) | 400 (3.4) | No inhibition |

**Fig. 4 Competitive binding assay results. a** OD$_{500}$ values with each concentration of Me-THF and 5 µM of *efm*SHMT in the presence and absence of each compound (black; No drug, red; (+)-SHIN-1, purple; PMX, green; MTX, yellow; SER and blue; SHMT-IN-2). Grey-, dark yellow- and light-coloured symbols show the values for calculating the average values and standard error (SE) of each point. The n number for No drug is ten (n = 3 at 8000 µM of Me-THF, n = 9 at 2000 and 4000 µM of Me-THF and n = 10 at other data points). The data point for SHMT-IN-2, SER, PMX and MTX is four (n = 4). The n number for (+)-SHIN-1 is five (n = 3 at µM of Me-THF, n = 4 at 2000 and 4000 µM of Me-THF and n = 5 at other data points). Addition of (+)-SHIN-1 and PMX suppressed Me-THF binding to *efm*SHMT. **b** $K_i$ of each compound and other values, which were used for calculating Ki, are shown. Stronger $K_i$ were observed at the lower *efm*SHMT concentrations used. All experiments were performed in the presence of 10 mM Gly and 100 µM Ser. These experiments were performed three to ten times. pEC$_{50}$, pConc and p$K_i$ are log-transformed EC$_{50}$, Conc. and $K_i$, respectively.

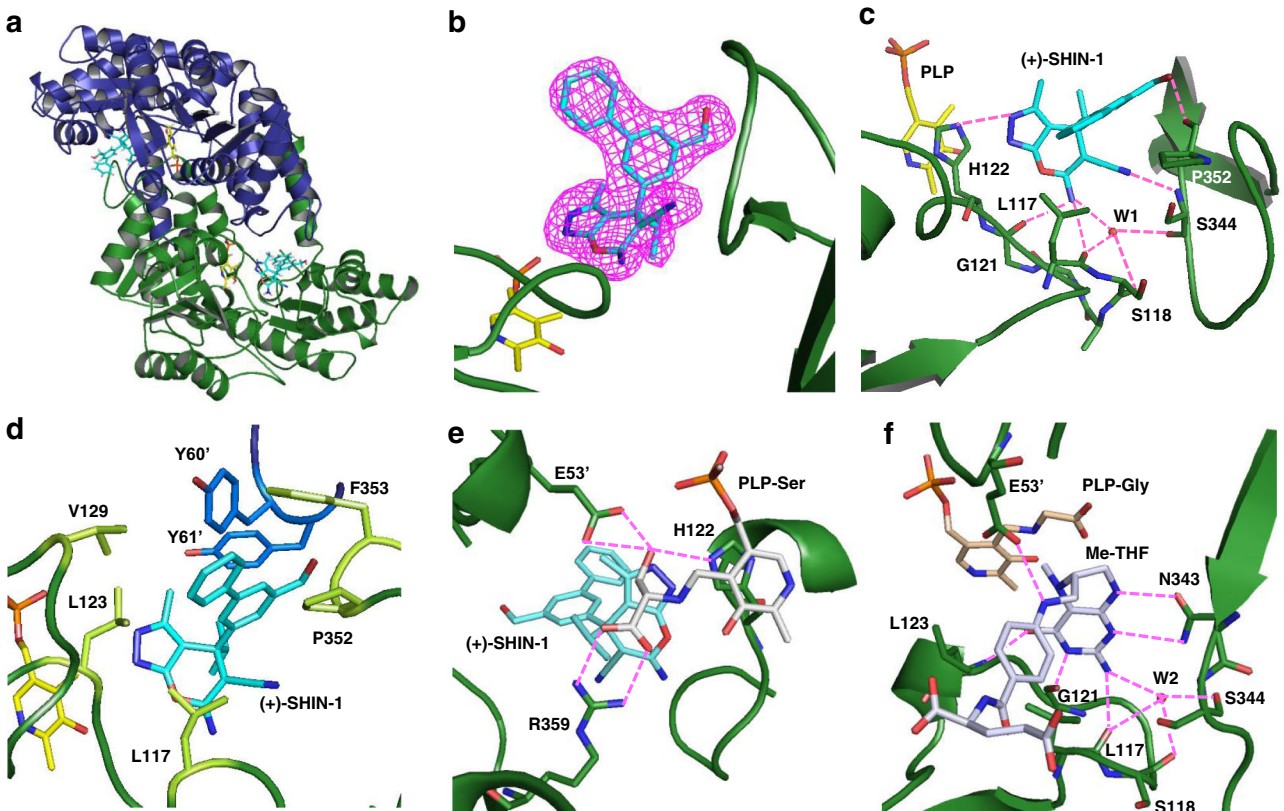

**Fig. 5 Crystal structure of the efmSHMT/(+)-SHIN-1 complex. a** Cartoon representation of the *efm*SHMT dimer. Green (chain A) and blue (chain B) cartoon structures are *efm*SHMT monomers. Yellow-coloured stick representation structures are PLP, whereas cyan-coloured stick representation structures are (+)-SHIN-1. **b** Electron density map of (+)-SHIN-1. The Fo–Fc map of (+)-SHIN-1 is contoured at 3.0 s. **c** Polar interaction between *efm*SHMT and (+)-SHIN-1. Magenta dotted lines are hydrogen bonds or polar interactions. Exocyclic amine and hydroxy groups in (+)-SHIN-1 form hydrogen bonds with the mainchain of G121, L117 and P352. The exocyclic amine group also forms water (W1)-mediated hydrogen bonds with S344, L117, and S118. The pyrazole moiety interacts with the H122 side chain via hydrogen bonding. The exocyclic cyano group also forms polar interactions with the S344 main chain. **d** Non-polar interactions between *efm*SHMT and (+)-SHIN-1. The biphenyl moiety in (+)-SHIN-1 interacts hydrophobically with L117, L123, V129, P352 and F353 in chain A. The biphenyl moiety also interacts with Y60′ and Y61′ in chain B. **e** Ser bound to PLP hydrogen bonds with H122 and R359 in chain A, and E53′ in chain B. **f** Polar interactions between *efm*SHMT and Me-THF. Me-THF interacts directly with L117, G121, L123 and N342 in chain A and E53′ in chain B. The exocyclic amine group also forms water (W2)-mediated hydrogen bonds with S344, L117 and S118.

**Table 3 Data collection and refinement statistics (molecular replacement).**

| | *efm*SHMT/ (+)-SHIN-1 | *efm*SHMT/Ser/ (+)-SHIN-1 | efmSHMT/ Gly/Me-THF |
|---|---|---|---|
| *Data collection* | | | |
| Space group | $P4_12_12$ | $P4_12_12$ | $P4_12_12$ |
| Cell dimensions | | | |
| *a, b, c* (Å) | 119.470, 119.470, 163.280 | 119.35, 119.35, 162.588 | 126.981, 126.981, 170.106 |
| a, b, g (°) | 90, 90, 90 | 90, 90, 90 | 90, 90, 90 |
| Resolution (Å) | 48.21–2.28 (2.362–2.28) | 49.35–1.90 (1.968–1.90) | 47.23–2.62 (2.714–2.62) |
| $R_{sym}$ | 0.437 (3.76) | 0.2563 (3.19) | 0.6466 (3.40) |
| *I / sI* | 8.12 (1.04) | 14.30 (1.00) | 13.66 (1.49) |
| Completeness (%) | 100 (100) | 99.94 (99.95) | 99.92 (99.71) |
| Redundancy | 16.8 (15.8) | 26.8 (27.8) | 27.0 (27.6) |
| *Refinement* | | | |
| Resolution (Å) | 2.28 | 1.90 | 2.62 |
| No. reflections | 54467 | 92693 | 42441 |
| $R_{work}$ / $R_{free}$ | 0.1750/ 0.2086 | 0.1742/0.1974 | 0.1618/0.2012 |
| No. atoms | | | |
| Protein | 6241 | 6266 | 6297 |
| Ligand/ion | 92 | 104 | 139 |
| Water | 368 | 535 | 197 |
| *B-factors* | | | |
| Protein | 41.8 | 39.0 | 53.4 |
| Ligand/ion | 39.8 | 36.6 | 58.5 |
| Water | 43.4 | 42.8 | 50.7 |
| R.m.s. deviations | | | |
| Bond lengths (Å) | 0.008 | 0.006 | 0.008 |
| Bond angles (°) | 1.06 | 1.00 | 1.01 |

Number of xtals for each structure should be noted in footnote. Values in parentheses are for highest-resolution shell.
Equations defining various *R*-values are standard and hence are no longer defined in the footnotes.
Ramachandran statistics should be in Methods section at the end of Refinement subsection.
Wavelength of data collection, temperature and beamline should all be in Methods section.

side chain of His122, respectively. Residues Gly120 to Leu123, located in the 115-loop, are well conserved in SHMTs, and His122 interacts with PLP. The structure of the 115-loop around Leu123 and His122 is similar among the SHMTs examined. The cyano group in (+)-SHIN-1 forms a polar interaction with the main chain of Ser344. The biphenyl moiety in (+)-SHIN-1 forms hydrophobic interactions with Leu117, Leu123, Val129, Pro352 and Phe353 (Fig. 5d). Moreover, the biphenyl moiety hydrophobically interacts with Tyr60 and Tyr61 in another monomer unit that forms the *efm*SHMT dimer and the exocyclic hydroxy group hydrogen bonds with the main chain of Pro352. The binding mode between the (+)-SHIN-1 pyrazolopyran core and *efm*SHMT is similar to that observed in the *pv*SHMT/(+)-85 complex. (+)-85 is a pyrazolopyran derivative (PDB ID: 5GVN)[13]. The pyrazolopyran core of (+)-85 forms polar interactions with *pv*SHMT using the pyrazole, exocyclic amino and cyano moieties. However, in the *hu*SHMT-2/SHMT-IN-2 complex, the SHMT-IN-2 pyrazolopyran core forms a hydrogen bond with Ser226, and the exocyclic amine interacts with Leu166 and Gly170 in human SHMT-2. In contrast, the cyano group and pyrazole moiety do not form hydrogen bonds with *hu*SHMT-2 (PDB ID: 5V7I)[14].

In the structure of the *efm*SHMT/Ser/(+)-SHIN-1 complex, the covalent bond formed by a Schiff's base relocated from K226 to the amino group of Ser (Fig. 5e). The Ser amino acid forms hydrogen bonds with E53' (prime denotes the neighbour

molecule) and H122 to stabilise the 115-loop, and the carbonyl group of Ser also forms a salt bridge with R359. Similarly, the structure of *efm*SHMT/Gly/Me-THF showed the Schiff's base between PLP and Gly (Fig. 5f). As mentioned above, *ec*SHMT and *efm*SHMT share structural similarity. The binding mode of folate (Me-THF in *efm*SHMT and formyl-THF in *ec*SHMT) is also conserved. In the structure of *efm*SHMT/Gly/Me-THF, Me-THF forms hydrogen bonds with the side chains of E53' and N343 and main chain amide groups of L117, G121 and L123 (Fig. 5f).

**Quantitation of mRNA for folate-binding proteins**. To clarify the folate cycle of *E. faecium*, we identified folate-binding proteins coded in the *E. faecium* genome (Genbank code: UFYJ01000001.1) and likely to be part of the folate cycle of *E. faecium* (Fig. 3a). As a result, *E. faecium* uses SHMT as a primary pathway to supply 1 C units and the glycine cleavage system (GCS) as a support pathway to produce $CH_2$-THF. Additionally, *E. faecium* was found to have a formyltetrahydrofolate synthetase (Fhs)[27], providing $CH_2$-THF by mediating FolD (Fig. 3a). The mRNA levels of the identified folate-binding proteins in the presence and absence of (+)-SHIN-1 were compared by real-time PCR (RT-PCR) to analyse the antibacterial mechanism of (+)-SHIN-1. The mRNA levels of SHMT and glycine cleavage system H protein (GcvH) were found to increase slightly 4 h after adding (+)-SHIN-1 to *E. faecium* cells in the growth phase (Fig. 2a). GCS also supplies $CH_2$-THF from THF and Gly. Fhs and cobalamin-independent methionine synthase (MetE) mRNA levels decreased slightly, whereas the mRNA of other folate-binding proteins did not change (Fig. 3b). The mRNA levels of folate-binding proteins, except TS and DHFR, increased 10 h after adding (+)-SHIN-1 to *E. faecium* cells in the stationary phase (Fig. 2a, Fig. 3b). In particular, the mRNA amount for the expression of TS decreased drastically. Thus, thymidine starvation occurred in *E. faecium* at this time point.

**Synergistic effects among 1C metabolism inhibitors including (+)-SHIN-1**. To investigate whether 5-FdU and (+)-SHIN-1 have synergistic antibacterial effects, we determined the $EC_{50}$ for these two inhibitors using various drug combinations and plotted isobolograms (Fig. 7)[28]. Data points falling within the lower left-hand side of an isobologram indicate a synergism effect. As shown in Fig. 7 and Supplementary Tables S4 to S6, (+)-SHIN-1 synergistically enhanced the antibacterial activity of 5-FdU and 5-FUrd. This observation indicates that the SHMT inhibitor synergistically enhanced the antibacterial activities of these compounds, which then inhibited DNA and RNA synthesis downstream of 1C metabolism. MTX also synergistically enhanced the antibacterial activity of 5-FdU and 5-FUrd. Additionally, combined (+)-SHIN-1 and MTX acted synergistically in bacterial cells. The synergistic effect of SHMT inhibition with MTX in T-cell acute lymphoblastic leukaemia has been reported[29].

**Discussion**

In this report, we determined that (+)-SHIN-1 functions as an SHMT inhibitor against *E. faecium* with extremely high potency and low cell cytotoxicity. (+)-SHIN-1 also exhibited synergism with 5-FdU and 5-FUrd. As previously reported, we found that 5-FdU blocks thymidine synthesis by competitively inhibiting deoxyuridine phosphatase and TS. 5-FUrd inhibited RNA synthesis as a uridine-kinase competitor. (+)-SHIN-1 inhibited SHMT and blocked the production of $CH_2$-THF, which is an essential biomolecule for thymidine and purine synthesis. Thus, in combination, (+)-SHIN-1 and these nucleoside analogues downregulated several pathways related to pyrimidine and purine

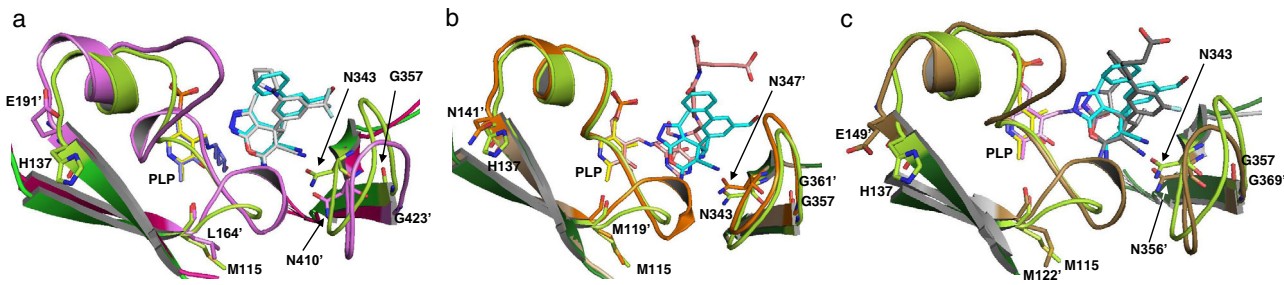

**Fig. 6 Comparison of the catalytic site of *efm*SHMT and three other SHMTs derived from human, *E. coli* and *P. vivax*.** Overlaid catalytic-site structures are shown in (**a–c**). **a** *efm*SHMT (green) and huSHMT2 (magenta), (**b**) *efm*SHMT (green) and ecSHMT (ivory), and (**c**) *efm*SHMT (green) and pvSHMT (silver) showing the divergence of the loop structures (light green, *efm*SHMT; pink, *hu*SHMT; orange, ecSHMT; gold, pvSHMT). Substrates bound to each SHMT are shown in stick representation (cyan, (+)-SHIN-1; white, SHMT-IN-2; beige, 5-formyl-H4PetGlu; dark grey, (+)-85). PLP binding to each SHMT is also shown in stick representation (yellow, *efm*SHMT; blue, *hu*SHMT; beige, ecSHMT; magenta, pvSHMT).

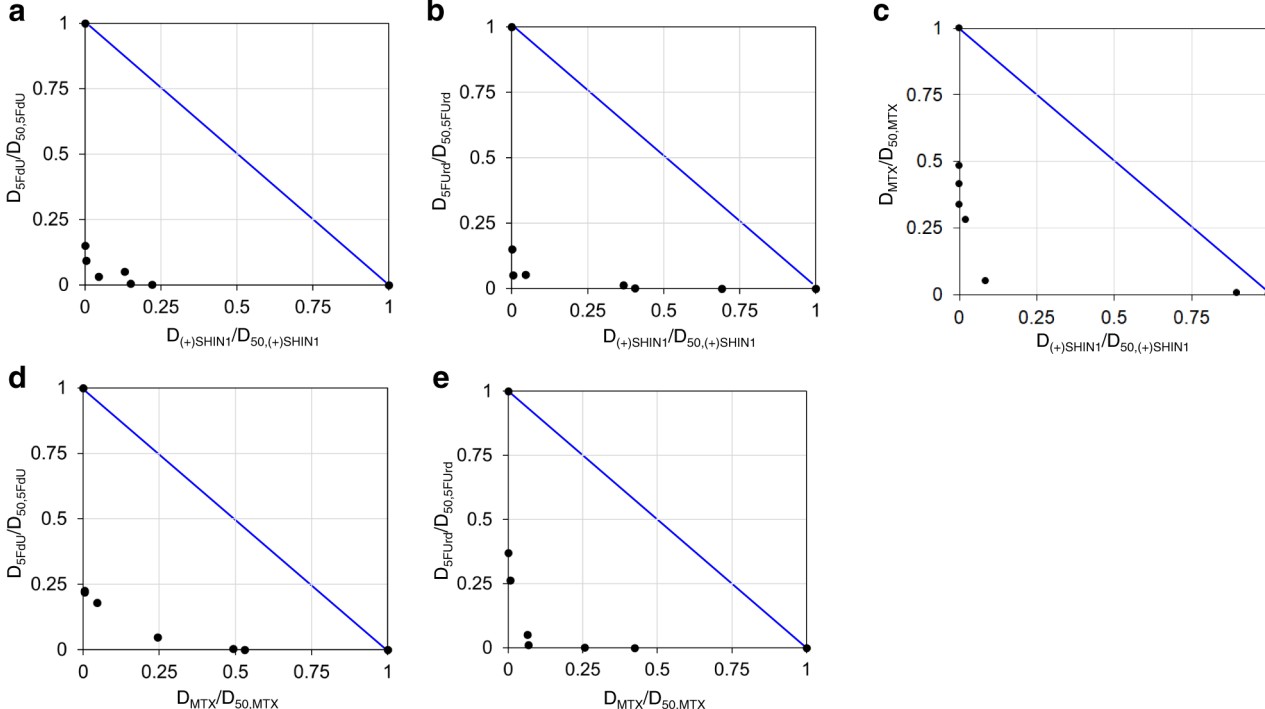

**Fig. 7 Synergistic effect of each drug combination. a–e** Isobolograms for each drug combination are shown. Synergistic effect of (**a**) (+)-SHIN-1 and 5-FdU, (**b**) (+)-SHIN-1 and 5-FUrd, (**c**) (+)-SHIN-1 and MTX, (**d**) MTX and 5-FdU, and (**e**) MTX and 5-FUrd. $D_A$ and $D_B$ are the concentrations of compounds A and B, respectively, that achieve a 50% effective concentration under a drug combination. $D_{50, A}$ and $D_{50, B}$ denote the EC$_{50}$ values under monotherapy shown in Supplementary Tables S5–7.

synthesis, thereby synergistically enhancing thymidine starvation and creating a nucleotide imbalance. Additionally, a single dose of (+)-SHIN-1 acted bacteriostatically and not as a bactericide. *E. faecium* utilises the supportive GCS and Fhs-FolD pathways to produce THF and CH$_2$-THF (Fig. 3a). The data suggested that (+)-SHIN-1 exerts a bacteriostatic rather than a bactericidal effect because *E. faecium* can recycle THF and CH$_2$-THF and prevent cell death by regulating the expression levels of folate-binding proteins, including GcvH and Fhs, as mRNA quantitation experiments showed.

Previously, Alfadhli et al. reported that *E. coli* mutants that lack SHMT were unable to synthesise glycine and failed to proliferate in the absence of glycine[30]. In a previous report, *E. coli* lacking SHMT was able to grow in the presence of excess Gly because CH$_2$-THF was supplied through GCS in the presence of excess Gly and also guanosine triphosphate (GTP) was provided

through the purine synthesis pathway. In contrast, we showed that *E. faecium* was unable to grow in the presence of excess Gly. These observations revealed differences in the folate synthesis pathway between *E. coli* and *E. faecium*. Thus, *E. coli* synthesised folate using GTP, but *E. faecium* lacked the pathway to provide GTP. Therefore, *E. faecium* cannot synthesise folate even in the presence of excess Gly, suggesting that *E. faecium* growth in the absence of folate is hampered. Moreover, (+)-SHIN-1 may inhibit SHMT and GCS-related proteins; however, the results from the competitive binding assay indicated that the main target of (+)-SHIN-1 is *efm*SHMT. Further research is required to clarify these possibilities.

The potent activity of (+)-SHIN-1 was only directed against *E. faecium* among the tested bacteria. *Enterococcus spp.* and *Lactobacillus spp.* require folate-related compounds for growth, and both can incorporate them[18]. The other microorganisms cannot

incorporate folate derivatives. Therefore, (+)-SHIN-1 would not be incorporated by other bacteria, as observed with DHFR inhibitors, suggesting that further improvements in the chemical structure of (+)-SHIN-1 are required for developing novel antibiotics that exhibit potent activities against a broad range of bacteria. Nevertheless, as shown using the competitive assay, the antibacterial mechanism underlying SHMT inhibition involves thymine and, in part, purine starvation. These antibacterial mechanisms share similarities with TMP, which exerts potent activities against Gram-positive and -negative bacteria. Furthermore, a previous report showed that SHIN-1 lacked pharmacokinetic properties suitable for the in vivo study of SHMT biology, but a SHIN-1 derivative, which acted as a human SHMT inhibitor, overcame these pharmacokinetic issues and exerted effective anticancer activity in vivo[29]. Therefore, SHMT inhibition is a potential target of novel antibiotics.

The crystal structure of the SHMT/(+)–SHIN-1 complex revealed that (+)-SHIN-1 binds strongly to efmSHIMT by forming several hydrogen bonds and hydrophobic interactions. The pyrazolopyran core of (+)-SHIN-1 interacts with the catalytic site of efmSHMT by forming hydrogen bonds with the main chain. A hydrogen bond formed with the main chain of Leu123, located in the 115-loop, is highly conserved among SHMTs. These hydrogen bonds should contribute to the high genetic barrier of (+)-SHIN-1. Darunavir, a protease inhibitor used for human immunodeficiency virus (HIV) infectious diseases, also forms strong hydrogen bonds with mainchain residues in the catalytic site of the HIV protease[23,31]. These hydrogen bonds contribute to the high genetic barrier of Darunavir[32]. Indeed, in vitro drug-resistance induction suggests that Darunavir prevents HIV-1 protease developing significant resistance using in vitro drug-resistance induction[33]. Thus, the pyrazolopyran core of (+)-SHIN-1 should prevent efmSHMT developing resistance by forming hydrogen bonds with the main chain of efmSHMT. Furthermore, the competitive binding assay showed that the binding affinity of (+)-SHIN-1 to efmSHMT was enhanced under coexistence with Ser when compared with Gly, while that of Me-THF did not change. In the structure of the efmSHMT/Ser/(+)-SHIN-1 ternary complex, the serine residue on PLP formed hydrogen bonds with E53 and H122. H122 is located on the 115-loop, and this hydrogen bond stabilises the loop. Structural analysis revealed that (+)-SHIN-1 forms several hydrogen bonds with amino acids in the 115-loop. Loop stabilisation contributes to the binding affinity of (+)-SHIN-1 to efmSHMT. In contrast, the binding affinity of Me-THF was not altered in the assay, as mentioned above. Me-THF also hydrogen bonds with amino acids in the 115-loop. Loop stabilisation can also contribute to the binding affinity of Me-THF. In the efmSHMT/Gly/Me-THF structure, E53 formed a hydrogen bond with Me-THF. Although the loss of the hydrogen bond with the 115-loop may negatively affect the binding affinity of Me-THF, the hydrogen bond with N10 positively contributes to the binding affinity. For Me-THF binding, each contribution was countered by hydrogen bond switching, and no change in the binding affinity was observed. Notably, we found that in E. faecium, (+)-SHIN-1 and SHMT-IN-2 exhibit quite distinct antibacterial mechanisms, and SHMT-IN-2 did not bind to efmSHMT despite both compounds having the same pyrazolopyran core. These two compounds have different substructures at the biphenyl moiety position in (+)-SHIN-1. Thus, the chemical structure at this position is clearly important for the specificity of SHMT. We found that the biphenyl moiety in (+)-SHIN-1 mainly interacts hydrophobically with the aa side chains in the 115- and 343-loops. The exocyclic hydroxy group in (+)-SHIN-1 clearly interacts with the 342-loop. As reported for pvSHMT, the structure of the 343-loop is highly flexible. Additionally, the 115-

and 343-loops share low sequence homology. The biphenyl and hydroxy groups in (+)-SHIN-1 stabilised efmSHMT/(+)-SHIN-1 binding by reducing the flexibility of the 343-loop. However, SHMT-IN-2 neither formed a hydrogen bond with the 343-loop nor bound to efmSHMT. Furthermore, Ser binding to efmSHMT enhanced the binding affinity of (+)-SHIN-1 by stabilising the 115-loop. However, there were no direct interactions between (+)-SHIN-1 and Ser complexed with efmSHMT. Thus, for the biphenyl moiety in (+)-SHIN-1, developing modifications that maintain the interaction with the 343-loop should generate novel, effective SHMT inhibitors that display species specificity. The modification using SHMT-IN-2 as a basic structure would facilitate the generation of huSHMT specific anticancer drugs, which do not affect intestinal flora. Furthermore, developing the pyrazolopyran moiety for direct interaction with Ser complexed with efmSHMT should yield more potent SHMT inhibitors than (+)-SHIN-1.

It was previously reported that PMX could bind to human SHMT-2 and Arabidopsis thaliana SHMT[25,34]. Thus, PMX could work as a species-wide SHMT inhibitor. This information is also useful to develop novel more potent and specific inhibitors of the emerging anti-bacterial, -cancer and -parasite drug target. On the other hand, in this study, it was proved that PMX bound to efmSHMT although the binding affinity was too weak to work as an efmSHMT inhibitor within the concentration range exerting antibacterial activity. Additionally, SHMT-IN-2 and SER, which are reported to be huSHMT inhibitors, never suppress SHMT activity as competitive binding assay and DSF proved. SER reportedly works as an efflux inhibitor against E. coli[35] and probably against E. faecium. SHMT-IN-2 also failed to bind to SHMT and did not compete against any competitor we tested. This observation suggests that SHMT-IN-2 will likely fail to inhibit SHMT and other enzymes involved in folate-mediated 1 C metabolism. Further research is required to determine the detailed mechanism underlying the antibacterial activity of SHMT-IN-2.

In our results, the SHMT inhibitor synergistically enhanced the antibacterial activities of nucleoside analogues. Hydroxyurea, which decreases purine triphosphate levels by inhibiting ribonucleotide reductase, reportedly enhances the type 1 anti-human immunodeficiency virus activity of azido-3′-deoxythymidine, 2′,3′-dideoxycytidine and, in particular, 2′,3′-dideoxyinosine[36–38]. SARS-CoV-2 is reported to induce purine synthesis, supporting massive viral subgenomic RNA by hijacking serine metabolism through human SHMT1[11]. Therefore, SHMT inhibitors might enhance the anti-SARS-CoV-2 activities of RNA-dependent RNA polymerase inhibitors such as remdesivir. Thus, SHMT inhibitors may also play important roles in combination therapies for bacterial and viral infections.

In conclusion, we have shown that SHMT is a novel antibacterial target in E. faecium. New antibiotic classes are urgently required to develop therapeutics that possess potent activities against multi-drug resistant bacteria and cannot induce cross-resistance to currently approved antibiotics. We found that (+)-SHIN-1 exerts strong antibacterial activity and combining (+)-SHIN-1 and nucleoside analogues produced strong synergistic effects. The crystal structure of the efmSHMT/(+)-SHIN-1 complex showed that the pyrazolopyran core of (+)-SHIN-1 forms several hydrogen bonds with the main chain of efmSHMT and that the biphenyl moiety and hydroxy group of (+)-SHIN-1 are important stabilisers of the interaction between efmSHMT and (+)-SHIN-1 because these moieties interact and reduce the flexibility of the 115- and 343-loops. Our findings should facilitate the development of SHMT inhibitors as therapeutic agents for bacterial, viral and parasite infections and for treating cancer.

## Materials and methods

**Compounds and bacterial strains.** SHIN-1, (+)-SHIN-1, PMX, MTX, nolatrexed, RTX and SER were purchased from MedChemExpress (NJ, USA). SHIN-1 is the mixture of two enantiomers, (+)-SHIN-1 and (−)-SHIN-1. (+)-SHIN-1 was reported as the active compound against SHMT in humans. SHMT-IN-2 was obtained from MedKoo Biosciences (NC, USA). 5-FdU, 5-FUrd and TMP were purchased from Tokyo Chemical Industry (Tokyo, Japan). Vancomycin was purchased from Wako (Osaka, Japan). *E. faecium* (JCM5804) and *E. faecalis* (JCM7783) were obtained from the Japan Collection of Microorganisms (JCM, Ibaraki, Japan). *S. aureus* (ATCC29213), *Escherichia coli* (ATCC25922) and Calu-3 cells were purchased from the American Type Culture Collection (ATCC, VA, USA). Hep2 cells were a kind gift from Fukushima Medical University. Caski cells were purchased from the Cell Resource Center for Biomedical Research (Cell Bank of Tohoku University).

**Antibacterial activity evaluations.** All bacterial strains were grown overnight at 37 °C on Nissui Plate Sheep Blood Agar (Nissui, Tokyo, Japan). *E. faecium* and *E. faecalis* were diluted to turbidity equivalents that matched 1.0 McFarland (McF) standard in distilled water (*S. aureus* and *E. coli* were also diluted to 0.5 McF), and additionally diluted 1:200 in Mueller–Hinton (MH) broth. The designated concentrations of the tested compounds were prepared by ten-fold serial dilutions directly in 96-well microtiter plates (IWAKI, Shizuoka, Japan), as previously described[19]. The maximal concentration of SHIN-1 and (+)-SHIN-1 was 4 μM each (final concentration, 2 μM), and TORAST-H tip (SHIMADZU, Kyoto, Japan) was used for the serial dilutions. The other compounds were serially diluted from 20 μM downwards (final concentration, 10 μM). MH-suspended bacterial aliquots (100 μL each) were added to 100 μL of MH containing 0.001–10 μM of each compound (0.2 pM–2 μM of SHIN-1 or (+)-SHIN-1) in 96-well microtiter plates. Positive control wells included bacteria without any compound, and the negative control wells only contained MH broth. Each experimental and control well had a final volume of 200 μL. Plates were incubated for 24 h (*S. aureus* and *E. coli* were incubated for 18 h) at 37 °C. The optical density at 600 nm (OD$_{600}$) was measured using a GloMax microtiter plate reader (Promega, WI, USA), and the EC$_{50}$ values of each test compound and standard error were calculated by Microsoft Excel function, AVERAGE and STDEV/SQRT.

**Determination of bacterial viability.** Living *E. faecium* were determined using the Microbial Viability Assay Kit-WST (DOJINDO, Tokyo, Japan). Briefly, *E. faecium* were grown overnight at 37 °C on Nissui Plate Sheep Blood Agar. *E. faecium* were diluted to a turbidity equivalent that matched 1.0 McF standard in distilled water, followed by a further dilution of 1:200 in MH broth with or without drugs. Positive control wells included bacteria seeded into MH broth without compounds, and the negative control wells contained MH broth only. All experimental and control wells had a final volume of 200 μL. Five microlitres of the colouring reagent were added to each well of the 96-well plate every 2 h after starting incubation with each compound. The absorbance at 450 nm (OD$_{450}$) was measured 1 h after adding the colouring reagent using the GloMax microtiter plate reader (Promega, Tokyo, Japan). EC$_{50}$ values were calculated using the values of relative bacterial viability after 14 h incubation.

**Cell cytotoxicity assays.** Calu-3 and Caski cells (100 μL, $1 \times 10^5$ cells/mL) were cultured in RPMI (Sigma-Aldrich, St Louis, MO) containing 10% foetal bovine serum (FBS; Thermo Fisher Scientific, MA, USA), 100 units/mL penicillin G and 50 μg/mL streptomycin at 37 °C in 5% CO$_2$. Hep2 cells (100 μL, $1 \times 10^5$ cells/mL) were cultured in DMEM (Sigma-Aldrich) containing 10% FBS, 100 units/mL penicillin G and 50 μg/mL streptomycin (DMEM-FBS) at 37 °C in 5% CO$_2$. Each inhibitor was added to the 96-well microtiter plates and serially diluted ten-fold to produce concentrations ranging from 0.01–10 μM using the FBS-containing medium added directly to the plates. After serial dilution, 100 μL of FBS-containing medium including the cells was added to each well of the plate. The 50% cytotoxicity concentration (CC$_{50}$) of each compound was assessed by determining the cell viability using Cell Counting Kit-8 (DOJINDO, Kumamoto, Japan) after the cells were incubated with the serially diluted compounds for 5 days. All CC$_{50}$ determining assays were conducted in duplicate or triplicate[39,40].

**Antibacterial assay with an excess concentration of biomolecules.** The EC$_{50}$ values of the test compounds against *E. faecium* were determined in the presence of different concentrations of Ser, Gly, inosine, dT and FA using the same procedure described for the test compounds. The ratio of the control EC$_{50}$ (no competitors added) to the experimental EC$_{50}$ (competitors added) was calculated for each test substance.

**Determining the synergistic effects of 5-FdU and (+)-SHIN-1.** The EC$_{50}$ values of (+)-SHIN-1, 5-FdU, 5-FUrd or MTX against *E. faecium* were determined in the presence of each concentration of each drug combination using the same procedure described above. The combination index (C.I.) was calculated using: C.I. = $D_A/D_{50, A} + D_B/D_{50, B}$, where $D_A$ and $D_B$ are the concentrations of compounds A and B, respectively, that achieve an EC$_{50}$ under drug combinations, and $D_{50, A}$ and $D_{50, B}$ represent the EC$_{50}$ values under monotherapy[28]. C.I. < 1.0 denotes a synergistic effect, C.I. = 1.0 denotes an additive effect and C.I. > 1.0 denotes an antagonistic effect.

**Plasmid construction.** *E. faecium* was diluted to a turbidity equivalent that matched 1.0 McF standard in distilled water. The diluted sample was heated at 98 °C for 10 min and then incubated at 4 °C for 5 min. The sample was centrifuged, and the supernatant was collected and used as a DNA template for PCR. *efm*SHMT DNA was PCR-amplified using the aforementioned DNA template and an appropriate primer set (*efm*SHMT-F1: 5′-TATATACCCGGGGTGGTAGATTAC AAAACGTTTGAC-3′ and *efm*SHMT-R1: 5′-TAATAGGATCCACACCTCATAT ACTAGAGAGCATCAC-3′). The amplified DNA fragment was purified using a PCR purification kit (QIAGEN, Venlo, the Netherlands) in accordance with the manufacturer's protocols. The DNA fragment and pET-47b (Merck, Darmstadt, Germany) were digested with *Xma*I (NEB, MA, USA) and *Bam*HI (TOYOBO, Shiga, Japan) at 37 °C for 2 h. The digested DNAs were purified using the same PCR purification kit and ligated using T4 DNA ligase (Promega) (room temperature, 2 h). The ligated sample was transformed into JM109 cells (Wako, Osaka, Japan). Transformed cells were spread onto kanamycin-containing lysogeny broth (LB) plates and incubated at 37 °C overnight. Fourteen hours later, the bacterial colonies were selected and incubated in LB, including kanamycin at 37 °C overnight. The incubated samples were harvested by centrifugation, and the SHMT expression vector extracted from the incubated bacteria using a miniprep kit (QIAGEN) was sequenced (Genetic Analyzer 3500).

**Protein expression and purification.** The *efm*SHMT and *ec*SHMT constructs with N-terminal histidine tags and PreScission protease cleavage sites were expressed in *E. coli* BL21(DE3) cells. The proteins were chromatographically purified on a HisTrap column (Cytiva, Tokyo, Japan) and digested with PreScission protease (Cytiva). The solutions were then subjected to HiTrap Q (Cytiva) and Superdex 75 gel filtration chromatography steps (Cytiva). Each protein preparation was concentrated in 20 mM HEPES buffer (pH 7.5) containing 50 mM NaCl (final concentration, 15 mg/ml) and stored at 4 °C.

**Thermal stability analysis using differential scanning fluorimetry.** Protein samples were prepared as 20 μL aliquots containing 0.1 mg/mL protein (*efm*SHMT/*ec*SHMT), 1 μL of the ligand in DMSO (final drug concentration, 50 μM) and 2% or 4% SYPRO Orange (Invitrogen, MA, USA). Measurements were taken using StepOnePlus real-time PCR (Thermo Fisher Scientific) (temperature increment, 1.8%; range, 20–95 °C). Melting points were evaluated as the half-point of the maximum intensity of each curve.

**Competitive binding assay.** Protein samples were prepared as 100 μL aliquots containing 2.5 or 5 μM *efm*SHMT, 10 mM Gly, 100 μM Ser and each concentration (1, 2.5, 5, 10, 20, 50, 100, 200, 500, 1000, 2000, 4000 and 8000 μM) of Me-THF with 20 mM HEPES (pH 7.5) and 50 mM NaCl. After adding Me-THF, the *efm*SHMT solution was incubated for 30 min at room temperature. The optical density at 500 and 600 nm (OD$_{500}$ and OD$_{600}$) were then measured using a nanodrop spectrophotometer[24,25]. OD$_{600}$ was used as the background. After detection, each concentration of inhibitor was added to the *efm*SHMT solution and incubated for 30 min at room temperature. OD$_{500}$ and OD$_{600}$ were measured. The relative OD$_{500}$ intensity was calculated as relative OD$_{500}$ = (OD$_{500}$ − OD$_{600}$)/($_{MAX}$OD$_{500}$ − OD$_{600}$), where $_{MAX}$OD$_{500}$ is the average OD$_{500}$ value in the presence of 1000, 2000 and 4000 μM Me-THF without inhibitors. EC$_{50}$ was defined as the concentration of Me-THF that gives 50% of *efm*SHMT bound to Me-THF, relative OD$_{500}$ intensity of 0.5. The binding inhibition constant ($K_i$) was estimated using the Cheng-Prusoff equation: $K_i$ = IC$_{50}$/([Me-THF]/[EC$_{50}$ without inhibitor] + 1), where IC$_{50}$ is the concentration of inhibitor where 50% of *efm*SHMT is bound to the inhibitor in the presence of Me-THF and [Me-THF] is the concentration of Me-THF, in which 50% of *efm*SHMT is bound to Me-THF in the presence of the inhibitor. Here, IC$_{50}$ is the concentration of inhibitor added to the *efm*SHMT solution and [Me-THF] is the EC$_{50}$ in the presence of the inhibitor concentration. [EC$_{50}$ without inhibitor] is the EC$_{50}$ of Me-THF in the absence of the inhibitor.

**Dynamic light scattering measurements.** Dynamic light scattering (DLS) was performed at 25 °C using Zetasizer NanoS instrumentation (Malvern Instruments, Worcestershire, UK). Sample concentrations were adjusted to 0.5 mg/mL. Measurements were conducted in triplicate. Data were analysed using the algorithms included in the Zetasizer Nano software.

**Crystallisation.** Purified *efm*SHMT was passed through Amicon Ultra-15 10 K centrifugal filter units (Millipore) to give a solution containing *efm*SHMT (15–26 mg/mL) in 20 mM HEPES and 50 mM NaCl. A 400-μM aliquot of (+)-SHIN-1 and 600-μM aliquots of SHMT-IN-2 and SER were used for crystallisation. After centrifugation, the supernatants were collected and subjected to crystallisation using the hanging drop vapour diffusion and sitting drop vapour diffusion methods. Crystal Screen Series, Index, Polyethylene Glycol/Ion Screen Kit Series (Hampton Research, CA, USA) and Wizard Crystallization Screen Series (Emerald BioSystems Inc., WA, USA) were used for the first screening to

determine the optimum crystallisation conditions. efmSHMT/(+)-SHIN-1 complexes formed in 0.1 M sodium cacodylate trihydrate (pH 6.4), 2.0 M ammonium sulfate, 0.2 M calcium chloride and 0.5 mM spermine, whereas efmSHMT/Ser/(+)-SHIN-1 ternary complexes were formed in 0.1 M HEPES (pH 7.5) and 1.4 M sodium citrate tribasic dihydrate. efmSHMT/Gly/Me-THF ternary complexes were formed in 0.1 M sodium cacodylate/hydrochloric acid (pH 6.5), 2 M ammonium sulfate and 0.2 M NaCl. efmSHMT crystals complexed with (+)-SHIN-1 were retrieved, immersed in a cryoprotective solution containing 20% glycerol reservoir solution and flash-frozen in liquid nitrogen.

**X-ray diffraction data collection and processing details**. Diffraction data of the efmSHMT/(+)-SHIN-1 complex were collected at SPring-8 BL32XU[41], and diffraction data of efmSHMT/Ser-(+)-SHIN-1 and efmSHMT/Gly/Me-THF complexes were collected at SPring-8 BL41XU[42]. All data were collected under cryo-conditions (100 K) using the automatic collection system ZOO[43] and processed by the automatic data processing software KAMO[44] and the CCP4[43–46] software suite. The structure of the efmSHMT/(+)-SHIN-1 complex was solved using the molecular replacement method with Phaser[47]. The coordinates of the Streptococcus thermophilus SHMT (Protein Data Bank (PDB) ID: 4WXB)[48] were used as the search model. The initial phase of the efmSHMT/Ser-(+)-SHIN-1 complex was obtained from the structure of the efmSHMT/(+)-SHIN-1 complex, whereas the structure of the efmSHMT/Gly/Me-THF complex was solved by the molecular replacement method with Phaser[47] using the efmSHMT/(+)-SHIN-1 complex structure as a search model because large changes in cell parameters were observed. The model was corrected and further refined using Phenix[49,50] and Coot[51]. Data collection and refinement statistics are summarised in Table 3. Structural models in figures were generated using PyMol (http://www.pymol.org).

**Relative quantitation of mRNA for expression of folate-binding proteins**. E. faecium was incubated at 37 °C overnight in 20 mL of MH broth with and without 2 μM of (+)-SHIN-1. Bacterial RNA was extracted using TRIzol™ Max™ Bacterial RNA Isolation Kit (Thermo Fisher Scientific, Tokyo, Japan) following the manufacturer's protocols. For the quantitative polymerase chain reaction (qPCR), the One-Step TB Green® PrimeScript PLUS RT-PCR Kit (TaKaRa, Kusatsu, Japan) was used and PCR was performed using the Thermal Cycler Dice Real Time System Lite (TaKaRa). The mRNA of D-alanine-D-alanine ligase (ddl) was used as a housekeeping RNA[52]. The primer set for qPCR is described in Supplementary Table S1. The PCR protocol used was: reverse transcription step at 40 °C for 5 min, an initial heating step at 95 °C for 10 s; 40 cycles of the amplification step consisting of a 95 °C denaturation period for 5 s, a 61 °C annealing (58 °C for TS, 60 °C for SHMT and 54 °C for ddl) and extension period for 30 s (1 min for ddl); and a melting phase from 65 to 95 °C, following the standard protocols of the TaKaRa Dice system. The cycle threshold ($C_t$) was calculated by the second derivative maximum method. The relative RNA amounts of the parental strain and mutants were evaluated using the $\Delta\Delta C_t$ method and ddl as the housekeeping RNA. Thus, the relative transcription level (RTL) was calculated using: $RTL = 2^{(-\Delta\Delta C_t)}$, $\Delta\Delta C_t = \Delta C_t^{SHIN-1} - \Delta C_t^{No\ drug}$, where $\Delta C_t$ represents the $C_t$ value of the RNA of each protein standardised by the $C_t$ value of the mRNA of ddl ($C_t^{ddl}$). Thus, $\Delta C_t$ was calculated as: $\Delta C_t = C_t - C_t^{ddl}$.

**Statistics and reproducibility**. Statistical analysis was performed using Microsoft Excel. The p-values were calculated by t test. Average values and standard errors in each experiment were calculated from at least three independent experiments.

**Reporting summary**. Further information on research design is available in the Nature Research Reporting Summary linked to this article.

## Data availability
The structures of the efmSHMT/(+)-SHIN-1, efmSHMT/Ser/(+)-SHIN-1 and efmSHMT/Gly/Me-THF complexes has been deposited in the RCSB PDB (https://www.rcsb.org/; PDB ID: 7V3D, 7X5N and 7X5O, respectively). Source Data are available in Supplementary Data 1–4. All other data are available from the corresponding author on reasonable request.

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

## Acknowledgements

The authors are grateful to Kanako Kobayashi and Mizue Saito for editorial help. We acknowledge support from the Biomedical Research Core of the Tohoku University Graduate School of Medicine. This research was partially supported by Platform Project for Supporting Drug Discovery and Life Science Research (Basis for Supporting Innovative Drug Discovery and Life Science Research (BINDS)) from AMED under Grant Number JP21am0101070 (support number 1947). We thank the Drs Kunio Hirata, Naoki Sakai and Yoshiaki Kawano at the BL32XU of SPring-8 with for their technical assistance with data collection and data processing. The diffraction experiment at SPring-8 BL41XU was performed with the approval of the Japan Synchrotron Radiation Research Institute (JASRI) (proposal no. 2021B2560). We thank the Edanz Group (https://jp.edanz.com/ac) for editing a draft of this manuscript. This work was supported in part by research grants from the Japan Society of the Promotion for Science (JSPS No. 16H05346, 18H02555) and by the Joint Usage/Research Center for Zoonosis Control, Hokkaido University, Japan.

## Author contributions

C.O., K.M., H.H. and E.N.K. conceived and designed the experiments. Y.M., A.N. and H.H. made protein crystals. H.H., K.M. and K.H. (Kazuya Hasegawa) solved crystal structures. Y.M., K.I., H.O., K.H. (Kazushige Hirata), M.U. and M.K. performed bacterial assay. S.S., A.N. and K.K. performed cell cytotoxicity assay. T.H., Y.Y., M.S. (Mikako Shirouzu), K.M., and H.H. performed biological assay. Y.M., M.S. (Mina Sasano), E.U., and H.H. analyzed the data. Y.M. K.M. and H.H. wrote the manuscript. H.H. and E.N.K. directed the teams.

## Competing interests

The authors declare no competing interests.
