## [Peer Review File · Communications Biology]

Reviewers' comments:

Reviewer #1 (Remarks to the Author):

The manuscript submitted by Makino et al reports a scientifically sound investigation on the effectiveness of SHIN1 and SHIN2 SHMT inhibitors as antibacterial agents, using *Enterococcus faecium* as model organism. The authors show that SHIN1 has high potency as antibacterial agent and bacterial selectivity. A synergistic effect was also observed between SHIN1 and one-carbon unit metabolism inhibitors. In my opinion the manuscript represents a novel and interesting contribution on the possibility to use specific SHMT inhibitors as antibacterial agents. However, I do not think that the manuscript, as it is, is very convincing on the fact that the observed effect of SHIN is actually due to SHMT inhibition.

ABSTRACT

Lane 31. In "CH₂-THF", "2" is written in subscript.

Lane 36. Please modify the sentence "(+)-SHIN-1 inhibited *E. faecium* at...." specifying that it inhibits growth.

Lane 37. The sentence "SHMT inhibition mainly induced...." Assumes that the observed effects are due to SHMT inhibition. I do not think that any direct evidence is presented in this respect (see below).

INTRODUCTION

Lane 50. Tetrahydrofolate is misspelled.

Lanes 53-54. There is something wrong in this sentence.

Lane 58. MTX is methotrexate, not methanolate.

Lane 64-66. These two sentences repeat a concept that has already been written a few lines above. SHMT does not catalyse the retroaldol cleavage of serine when THF is present.

Lane 71. "...to support massive viral subgenomic RNA.." do you mean to support the synthesis of RNA?

RESULTS

Lane 326 "To evaluate the potency of SHMT as an antibacterial target,...." How can you be sure that the observed of the antibacterial effect of SHIN1 is specifically due to SHMT inhibition?

Lane 238. Please explain the difference between SHIN-1 and (+)-SHIN-1.

Lane 251. How do you explain the observed high bacterial selectivity when it is known that SHIN1 is a potent inhibitor of human SHMT. Could it be that multiple bacterial targets of SHIN are present? I mean a bacterial target other than SHMT?

Lane 256. I wouldn't call these experiments competition experiments. The effect of serine and glycine is rather a rescue effect, isn't it?

Lane 263 Please explain here why you think that these data suggest that efmSHMT is the target protein of SHIN1. In my view, this cannot be taken for granted. What do you intend with "as a minimum"?

Lanes 267-268. Why should serine compete with SHIN? I mean serine is the amino acid substrate whereas SHIN is a THF analogue. They are not expected to compete since they bind at two different sites.

Lanes 277-279. Couldn't it simply be that that SHIN1 and SER are less potent inhibitors of SHMT?

Lane 281. Please indicate the concentration of the compounds used in the DSF assays. You should reformulate saying that at that given concentration some of the compounds do not bind. I think that you should measure the dissociation constant of SHIN1 equilibrium binding, using SHMT from various sources. This may be done by fluorimetry and is a very important information.

Lane 304. Reference to Figure 3 is wrong.

Reviewer #2 (Remarks to the Author):

Makino and coworkers present a well-written manuscript detailing the antimicrobial effects of the SHMT inhibitor (+)-SHIN, a compound originally described as a human SHMT inhibitor for treatment of cancer. In light of the urgent need for novel antibiotics with activity against resistant bacterial strains this is important research. The authors show with relevant experiments that (+)-

SHIN exerts its extremely potent activity in bacteria by direct targeting of SHMT. The manuscript fits well within the scope of the journal. The reviewer recommends the manuscript for publication albeit with some issues that need to be addressed.

In humans the production of thymidine can be sustained by both SHMT1 and MTHFD1 enzymes, as they both can generate the methylene-THF substrate for TYMS. The review wonders if FoID, the bacterial homologue of MTHFD1 can compensate for SHMT inhibition in bacteria, and encourage the authors to elaborate on this. If FoID has relevance it should be added to Figure 2.

From all the data provided on sertraline it seems clear that this compound is not an SHMT inhibitor at all, and in fact the data provided to prove this in the original paper by Geeraerts et al (ref. 15) is very weak to begin with. The authors should make a stronger statement to this effect.

The activity data in tables 1, 2, 3 and supplementary tables 3, 4 and 5 are expressed as mean EC50s with standard deviations. While this is very common practice, it is scientifically incorrect to indicate spread of data on non-transformed EC50 with standard deviations, due to the logarithmic nature of the equations used to calculate the EC50 values. Instead, standard deviations should be used with log-transformed EC50s (pEC50). Alternatively data spread on non-transformed values can be indicated by confidence interval, e.g. 95% confidence interval (CI95). In addition, reporting non-transformed EC50s with SD could be an indication that the mean values have been calculated incorrectly also. Here the geometric mean should be used, while for log-transformed EC50 the 'normal' arithmetic mean should be used. Taken together, the authors should report pEC50 values with standard deviations. These are also more easy to compare.

Minor comments:

Line 1: change 'potent' to 'potential'.

Line 53: change 'supplies' to 'converts'.

Line 58: change 'methanolate' to 'methotrexate'.

Line 59: capitalize 5-fluorouracil.

Line 60: change 'agent classes' to 'agents'.

Line 82: if *E. faecium* can grow in folate-free medium how would SHMT inhibition have any effect, since it uses folate as substrate/cofactor?

Line 110: 'Hep2 was a kind gift'.

Line 129: How were the EC50 values calculated, which equation and software were used?

Line 135: remove space after Aldrich.

Line 177: remove dash between histidine and tags.

Line 182: at which temperature were the protein aliquots stored?

Line 201: remove dash between hanging and drop.

Lines 220-225: this paragraph seems to be a duplication from the paragraph above.

Line 228: change 'numbers' to 'number'.

Line 242: remove dash between 100 and μM .

Line 253: change 14000 to 14,000.

Line 259: remove 'those'.

Line 268: change 'conflict with' to 'counteract'.

Lines 281-290: the authors should comment on the biphasic nature of the melting curve induced by (+)-SHIN1, as this complicated calculating the melting point. This could be due to monomer/dimer melting.

Line 285: change 'never thermally stabilized' to 'did not significantly stabilize'.

Line 287: change 'might bind' to 'binds'.

Line 293: add 'in complex' before 'with' pyridoxal'.

Line 302: remove dash between binding and site.

Line 306: change 'looped' to 'loop'.

Line 379: change 'means' to suggests'.

Line 385: change 'possibly' to 'clearly'.

Line 386: remove 'substrate'.

Line 387: change 'side-chain' to side-chains'.

Lines 394-397: PMX also binds to human SHMT2 (see PubMed ID 31127856), which should be mentioned to strengthen the observation of PMX being a species-wide SHMT inhibitor.

Line 584: (C) should be (D).

Line 595: change 'was' to 'are'. Which isoform of huSHMT is shown, 1 or 2?

Line 598: change 'showed' to 'shown'.

Line 846: remove after (+)-SHIN1.

Reviewer #3 (Remarks to the Author):

The manuscript by Makino et al. reports an investigation of the anti-bacterial inhibition activities of SHIN-1 against *Enterococcus faecium*. The roles of SHIN-1 (in combination with other inhibitors/ligands) against serine hydroxymethyl transferase (SHMT) from *Enterococcus faecium* (efmSHMT) were identified. Key data reported are the crystal structure of efmSHMT in complex with SHIN-1.

Issues of concerns are listed below:

1. In bacterial inhibition assays, how can the authors be certain that those 1C metabolism inhibitors target specifically only SHMT? These inhibitors are antifolate agents which could also interfere with other folate enzymes. The molecular mechanism of bacterial inhibition is still unclear.
2. I am curious about the sensitivity and accuracy of spectrophotometric measurement to quantitate the bacterial cell growth and to evaluate the EC50. Measurement of turbidity is not very sensitive and could be limited at high absorbivity. The assays should also be done by measuring colony forming unit (CFU) as well.
3. The manuscript lacks comprehensive studies of steady-state and inhibition kinetics of the recombinant enzyme. A lot of the data presented are indirect.
4. ITC experiments can be used to evaluate direct interactions between the enzyme and ligands instead of using indirect ΔT_m values.
5. In competition assays, concentrations of competing substances used in the assays are rather high. Are these relevant to the physiology concentrations?
6. The authors should validate that efmSHMT is essential for efm cell growth. The deletion and functional complementation analysis should be investigated or discussed.
7. In addition to cell cytotoxicity, the authors should present in vivo pharmacokinetics and pharmacodynamics of (+)-SHIN-1 to evaluate dosage and safety used of (+)-SHIN-1.
8. Although (+)-SHIN-1 shows promising results as a potent inhibitor, further modification of (+)-SHIN-1 should result in more potent inhibitors. This aspect should be discussed.
9. The data would be clearer if the enzyme-inhibitor complex can be co-crystallized in the presence of substrate/product i.e. L-ser, Gly, THF, MTHF. Without having these complexes, the authors can compare the current structure with previously solved co-complex structures with substrates. This way, discussion on mode of inhibition can be more insightful.
10. The authors presented the synergistic effects of (+)-SHIN-1 and other antifolates i.e. 5-FdU and MTX. I am curious how these inhibitors interact with the target enzyme.
11. Besides X-ray structures presented, the authors should perform MD simulations to investigate the motion of (+)-SHIN-1 binding compared to other pyrazolopyran derivatives in other SHMT complexes.

12. The authors should perform the quantitative flux analysis of (+)-SHIN-1 to ensure that this compound does directly target to only SHMT.

13. Anything known about the mRNA transcription of efmSHMT? Could (+)-SHIN-1 affect the efmSHMT gene expression level?

March 10th, 2022

Re: COMMSBIO-21-2371-T (Serine hydroxymethyltransferase as a potent target of antibacterial agents acting synergistically with one-carbon metabolism-related inhibitors)

Dear Reviewers,

Please find the attached files of our revised manuscript entitled “Serine hydroxymethyltransferase as a potential target of antibacterial agents acting synergistically with one-carbon metabolism-related inhibitors”, co-authored by Yuko Makino *et al.* We are pleased to have received favourable reviews from you and the three reviewers. We are also thankful for the helpful comments made by the three reviewers. We provide point-by-point responses to the comments raised by the reviewers (see below).

Additionally, we have added Kazuya Hasegawa as an additional co-author. He analysed two new crystal structures of *efm*SHMT in complex with ligands. Furthermore, Medkoo, from which we purchased SHIN-2, renamed this compound SHMT-IN-2. Therefore, we have changed all SHIN-2 entries in the original manuscript to SHMT-IN-2.

Responses to Reviewer #1 comments

We appreciate the comment by Reviewer #1 that our research represents a novel and interesting contribution to the possible use of specific SHMT inhibitors as antibacterial agents.

ABSTRACT

1) Lane 31. In “CH₂-THF”, “2” is written in subscript.

We have changed “CH₂-THF” to “CH₂-THF” (line 31 in the revised manuscript).

2) Lane 36. Please modify the sentence “(+)-SHIN-1 inhibited *E. faecium* at.....” specifying that it inhibits growth.

We have rewritten the sentence as follows

The changed sentence in the Abstract section of the revised manuscript (lines 36–37).

BEFORE: “(+)-SHIN-1 inhibited *E. faecium* at a 50% effective concentration of 10⁻¹¹ M.”

AFTER: “(+)-SHIN-1 inhibited the growth of *E. faecium* at a 50% effective concentration of 10⁻¹¹ to 10⁻¹⁰ M.”

3) Lane 37. The sentence “SHMT inhibition mainly induced....” Assumes that the observed effects are due to SHMT inhibition. I do not think that any direct evidence is presented in this respect (see below).

We appreciate the comment raised by Reviewer #1. We have used a competitive binding assay to determine the binding affinity of (+)-SHIN-1 against *efm*SHMT (please see our response to comment No. 16, page 5 to 7). The results of this assay revealed that (+)-SHIN-1 binds to *efm*SHMT with high affinity. As Reviewer #1 commented, there is no direct evidence that SHMT inhibition induces mainly thymine and partial purine starvation. Nonetheless, in the presence of excess amounts of dT, the antibacterial activity of (+)-SHIN-1 was clearly lost. This result is similar to that observed for 5-FdU, which induces thymine-less death. However, in the presence of inosine, the antibacterial activity of (+)-SHIN-1 only decreased marginally. Additionally, mRNA quantitation indicated that *E. faecium* has supporting pathways to produce THF and CH₂-THF, and the mRNA amount of purine synthesis-related enzymes, PurN and PurH, increased 10 h after adding (+)-SHIN-1 (please see our

response to comment No. 18, page 8). Ten hours after adding (+)-SHIN-1, purine synthesis was enhanced; however, bacterial growth was suppressed, as shown in the bacterial viability assay (please see our response to comment No. 55, page 15). Therefore, purines may be synthesized, but the amount is insufficient to enable *E. faecium* growth. Furthermore, mRNA level of TS drastically decreased in the presence of (+)-SHIN-1, indicating thymidine starvation. These data suggested that (+)-SHIN-1 exerts a bacteriostatic rather than a bactericidal effect because *E. faecium* can recycle THF and CH₂-THF and prevent cell death by regulating the expression levels of these folate-binding proteins, including GcvH and Fhs, as mRNA quantitation experiments showed. Taking into consideration these observations, we have included text in the Abstract, Result and the Discussion section of the revised manuscript as follows.

NOTION: The Abstract and Discussion section of the revised manuscript has been changed significantly (Page 25 to 28 in this response letter). The Abstract before and after revision are presented on pages 28 of this response letter.

Additional sentences in the Revised section of the revised manuscript.

Please see our response for comment No. 18, Page 8 and 9.

INTRODUCTION

4) Lane 50. *Tetrahydrofolate* is misspelled.

We have changed 5,10-methylene-tetrahydroxyfolate to 5,10-methylenetetrahydrofolate.

5) Lanes 53-54. *There is something wrong in this sentence.*

Lane 52-54 in the revised manuscript. We have changed the sentence as follows:

BEFORE: In addition, TS supplies thymidine monophosphate (dT-MP) to deoxyuridine monophosphate (dU-MP) by transferring the 1C unit from CH₂-THF

AFTER: In addition, TS converts deoxyuridine monophosphate to thymidine monophosphate (dTMP) by transferring the 1C unit from CH₂-THF

6) Lane 58. *MTX is methotrexate, not methanolate.*

Lane 57 in the revised manuscript. We have changed “methanolate” into “methotrexate”.

7) Lane 64-66. *These two sentences repeat a concept that has already been written a few lines above. SHMT does not catalyse the retroaldol cleavage of serine when THF is present.*

We thank reviewer #1 for raising this comment. We have deleted the repetitious text and changed those sentences as follows (Please see lane 65 to 68, page 3 in the revised manuscript.):

BEFORE: SHMT, a pyridoxal 5'-phosphate (PLP)-dependent enzyme, catalyses the retro-aldol cleavage of serine to glycine and converts THF into CH₂-THF. The glycine and CH₂-THF supplied by SHMT are essential for pyrimidine, purine and amino acid syntheses. Therefore, drugs targeting SHMT appear to be advantaged by blocking the multi-metabolic pathways downstream of 1C metabolism (e.g., DNA and RNA synthesis), suggesting that SHMT is a potentially useful target of therapeutic agents directed at a variety of viruses, bacteria and parasite infections, and cancer, too.

AFTER: Therefore, drugs targeting SHMT can block multiple metabolic pathways downstream

of 1C metabolism (e.g., DNA and RNA synthesis), indicating that SHMT is a potential target of therapeutic agents directed at various viruses, bacteria and parasite infections, as well as cancer.

8) Lane 71. “.. to support massive viral subgenomic RNA..” do you mean to support the synthesis of RNA?

We thank reviewer #1 for this comment. The relationship between SARS-CoV-2 and one-carbon metabolism was reported by Zhang *et al.* In that report, how SARS-CoV-2 induces host cell nucleotide metabolism was not fully characterized. However, de novo purine synthesis was enhanced in SARS-CoV-2 infected cells, and SHMT1 inhibition reduced the infectious virus titre by 1-log, diminished +strand viral genomic RNA and nucleoprotein levels, and induced resistance to the viral cytopathic effect. We have changed the sentence to clarify these points, in accord with the comment made by reviewer #1.

Changed sentence in the Introduction section of the revised manuscript. (Lane 68 to 70, Page 3)

BEFORE: Indeed, it has been reported that severe acute respiratory syndrome coronavirus 2 (SARS-CoV-2) induces purine synthesis to support massive viral subgenomic RNA by hijacking serine metabolism through human SHMT1.

AFTER: Indeed, it has been reported that severe acute respiratory syndrome coronavirus 2 (SARS-CoV-2) induces purine synthesis to support the synthesis of viral subgenomic RNA by hijacking serine metabolism through human SHMT1.

RESULTS

9) Lane 326 “To evaluate the potency of SHMT as an antibacterial target,....” How can you be sure that the observed of the antibacterial effect of SHIN-1 is specifically due to SHMT inhibition?

We appreciate the important comment made by Reviewer #1. Because (+)-SHIN-1 is located at the folate binding site of *efmSHMT* in the crystal structure (Fig. 6e and 6f of the revised manuscript and see page 24 of this response letter), it is necessary to consider the possibility that (+)-SHIN-1 functions as a promiscuous inhibitor against other folate binding proteins in *E. faecium*. Nonetheless, it is challenging to comprehensively determine the interaction between (+)-SHIN-1 and all folate binding proteins. However, an additional experiment carried out showed that (+)-SHIN-1 binds *efmSHMT* with a K_i lower than 9.1×10^{-9} M, indicating a strong interaction between (+)-SHIN-1 and *efmSHMT* (see our response to comment No. 16 of reviewer #1, page 5 to 7 of this response letter). Furthermore, in this research, we compared two compounds with very similar structures, (+)-SHIN-1 and SHMT-IN-2 (see Fig. 1 of the revised manuscript). These two compounds have been reported to inhibit human SHMT (ref). Our results showed that SHMT-IN-2 did not bind to *efmSHMT*. Moreover, no folate binding protein recognized SHMT-IN-2, as shown in the competitive folate assay, suggesting that the chemical structures of the SHIN series cannot bind promiscuously to folate binding proteins and (+)-SHIN-1 selectively binds to *efmSHMT*. These data indicated that the main target of (+)-SHIN-1 was SHMT in *E. faecium*, as observed in humans

NOTE: The Discussion section of the revised manuscript has been changed significantly. The revised Discussion section is presented on pages 25 to 27 of this response letter.

10) Lane 238. Please explain the difference between SHIN-1 and (+)-SHIN-1.

SHIN-1 is a mixture of two enantiomers, (+)-SHIN-1 and (–)-SHIN-1. (+)-SHIN-1 was reported

to be the active compound against SHMT in humans. We have explained this in the Materials and Methods section of the revised manuscript.

Additional sentences in the Materials and Methods section of the revised manuscript. (Lane 102 to 103, Page 5 in the revised manuscript)

SHIN-1 is a mixture of two enantiomers, (+)-SHIN-1 and (–)-SHIN-1. (+)-SHIN-1 was reported to be the active compound against SHMT in humans.

11) Lane 251. *How do you explain the observed high bacterial selectivity when it is known that SHIN1 is a potent inhibitor of human SHMT. Could it be that multiple bacterial targets of SHIN are present? I mean a bacterial target other than SHMT?*

We appreciate the comment made by reviewer #1. As the crystal structures and antiviral assay with an excess folate concentration indicated, (+)-SHIN-1 was recognized as a folate derivative in *E. faecium*. Therefore, the high bacterial selectivity of (+)-SHIN-1 was derived from the difference by each bacterium to incorporate folate derivatives, as mentioned in the Discussion section of the manuscript. However, as we described in the responses to comments No. 9 and 16, the main target of (+)-SHIN-1 is SHMT in *E. faecium*, as observed for human SHMT. Please see pages 3 and 5 to 7 in this response letter.

12) Lane 256. *I wouldn't call these experiments competition experiments. The effect of serine and glycine is rather a rescue effect, isn't it?*

We appreciate the comment made by Reviewer #1. “Competition assay” was re-named to “Antibacterial assay with an excess concentration of biomolecules”. The additional X-ray structures determined showed that Ser and Gly do not compete with (+)-SHIN-1 for binding to *efmSHMT*. Thus, the binding sites of (+)-SHIN-1 and Ser on *efmSHMT* do not overlap. Furthermore, Ser enhanced the binding affinity of (+)-SHIN-1 toward *efmSHMT*, as revealed by the competitive binding assay. Interestingly, Ser and Gly also affected the growth of *E. faecium*, as shown in Supplementary Figs. S1 and S2 (page 21 in this response letter). The bacterial response in the presence of excess amounts of Ser and Gly is complex. Therefore, we did not investigate why growth was affected by these amino acids in this study. However, in the presence of 100 mM Ser, the antibacterial activity of (+)-SHIN-1 was weaker when compared with that observed in the absence of Ser. We have described these points in the Results section of the revised manuscript as follows.

Additional sentences in the Results section of the revised manuscript. (Line 308 to 312, Page 10 and 11)

The bacterial viability assay showed that *E. faecium* growth was affected in the presence of 100 mM Ser (Supplementary Fig. S1) and that the antibacterial activity of (+)-SHIN-1 with 100 mM Ser was weaker when compared with the antibacterial activity of (+)-SHIN-1 in the absence of Ser. The decrease in antibacterial activity with Ser was not caused by competition between (+)-SHIN-1 and Ser.

13) Lane 263 *Please explain here why you think that these data suggest that efmSHMT is the target protein of SHIN1. In my view, this cannot be taken for granted. What do you intend with “as a minimum”?*

We thank Reviewer #1 for their valuable comment. As reviewers #1 and #2 comment, we have

added a paragraph about the competitive binding assay in the Results section of the revised manuscript (please see comment No. 16, page 5 to 7 in this response letter). The binding assay results were clearly consistent with the DSF results, showing that (+)-SHIN-1 binds to *efmSHMT* with high affinity, indicating that the main target of (+)-SHIN-1 is SHMT in *E. faecium*, as observed in humans. Therefore, we deleted “These data suggest that, as a minimum, the target protein of (+)-SHIN-1 is *efmSHMT*.” and included a sentence in the paragraph for the binding assay of the revised Results section as shown in our response for the comment made by reviewer #1 (comment No. 16, page 4 in this response letter).

Deleted sentence

These data suggest that, as a minimum, the target protein of (+)-SHIN-1 is *efmSHMT*.

14) Lanes 267-268. Why should serine compete with SHIN? I mean serine is the amino acid substrate whereas SHIN is a THF analogue. They are not expected to compete since they bind at two different sites.

We thank reviewer #1 for this comment. The additional crystal structure and binding assay indicated that Ser enhanced the binding affinity between SHMT and (+)-SHIN-1. However, the bacterial viability assay showed that excess Ser suppressed bacterial growth in the absence or presence of (+)-SHIN-1 (Please see our response for comment No. 12, page 4 and 21 in this response letter and Supplementary Fig. S1 in the revised manuscript). The suppression by excess Ser affected the EC₅₀ of (+)-SHIN-1.

15) Lanes 277-279. Couldn't it simply be that that SHIN2 and SER are less potent inhibitors of SHMT?

We thank Reviewer #1 for this comment. We have simplified the text by deleting and changing sentences as follows.

The deleted sentence.

“These data also indicate that SHMT-IN-2 and SER each have a mechanism distinct from that of (+)-SHIN-1 in *E. faecium*, despite these compounds working as human SHMT inhibitors.”

Changed sentences in the Results section of the revised manuscript. (Lane 328, Page 11)

BEFORE: Thus, SHMT-IN-2 and SER did not target the folate cycle-related metabolic enzymes.

AFTER: Thus, SHMT-IN-2 and SER did not target the folate cycle-related metabolic enzymes including SHMT.

16) Lane 281. Please indicate the concentration of the compounds used in the DSF assays. You should reformulate saying that at that given concentration some of the compounds do not bind. I think that you should measure the dissociation constant of SHIN1 equilibrium binding, using SHMT from various sources. This may be done by fluorimetry and is a very important information.

We appreciate the very important comment made by Reviewer #1. Direct binding of compounds to *efmSHMT* was evaluated by a binding assay that detected changes in absorbance at 500 nm, which is derived from quinonoid formation in the ternary complex of SHMT, Gly and 5-methyltetrahydrofolate (Me-THF) (1, 2). In the presence of excess Gly, the *efmSHMT*-Me-THF complex forms quinonoid, whereas the *efmSHMT*-inhibitor complex does not form quinonoid. Therefore, the addition of a compound to the ternary (*efmSHMT*-Gly-Me-THF) complex decreases the absorbance at 500 nm (OD₅₀₀) derived from quinonoid if the compound acts as an SHMT inhibitor (see page 23 in this response letter, Fig. 5 in the revised manuscript and Supplementary

Figs. S4 and S5 in the revised Supplementary Information). Using this feature and the Cheng-Prusoff equation, we estimated the binding inhibition constant (K_i) of compounds. The results of the binding assay revealed that the K_i of (+)-SHIN-1 is 0.0091 μM in the presence of 2.5 μM *efmSHMT* (the detection limit concentration). The Cheng-Prusoff equation assumes that the protein concentration is sufficiently low. However, in our assay, the detection limit concentration of *efmSHMT* was too high to calculate K_i of (+)-SHIN-1 accurately because the binding affinity of (+)-SHIN-1 was too strong. Thus, the actual K_i value of (+)-SHIN-1 is estimated to be lower than 0.0091 μM . Indeed, lowering the *efmSHMT* concentration resulted in lower K_i values for (+)-SHIN-1. Additionally, we also showed that PMX binds to *efmSHMT*; however, the binding affinity of PMX ($K_i = 123 \mu\text{M}$) was much weaker than that of (+)-SHIN-1. SHMT-IN-2, MTX and SER did not change the OD₅₀₀, indicating that these compounds did not function as SHMT inhibitors. These results were consistent with the DSF results. For binding assay, we have added sentences as shown below.

Additional sentences in the Materials and Methods section of the Revised manuscript. (Lane 199 to 216, Page 7 and 8)

Competitive binding assay. Protein samples were prepared as 100 μL aliquots containing 2.5 or 5 μM *efmSHMT*, 10 mM Gly, 100 μM Ser and each concentration (1, 2.5, 5, 10, 20, 50, 100, 200, 500, 1000, 2000, 4000 and 8000 μM) of 5-methyltetrahydrofolate (Me-THF) with 20 mM HEPES (pH 7.5) and 50 mM NaCl. After adding Me-THF, the *efmSHMT* solution was incubated for 30 min at room temperature. The optical density at 500 and 600 nm (OD₅₀₀ and OD₆₀₀) were then measured using a nanodrop spectrophotometer. OD₆₀₀ was used as the background. After detection, each concentration of inhibitor was added to the *efmSHMT* solution and incubated for 30 min at room temperature. OD₅₀₀ and OD₆₀₀ were measured. The relative OD₅₀₀ intensity was calculated as: $\text{relative OD}_{500} = (\text{OD}_{500} - \text{OD}_{600}) / (\text{MAXOD}_{500} - \text{OD}_{600})$, where MAXOD_{500} is the average OD₅₀₀ value in the presence of 1000, 2000 and 4000 μM Me-THF without inhibitors. EC₅₀ was defined as the concentration of Me-THF that gives 50% of *efmSHMT* bound to Me-THF, relative OD₅₀₀ intensity of 0.5. The binding inhibition constant (K_i) was estimated using the Cheng-Prusoff equation: $K_i = \text{IC}_{50} / ([\text{Me-THF}] / [\text{EC}_{50} \text{ without inhibitor}] + 1)$, where IC₅₀ is the concentration of inhibitor where 50% of *efmSHMT* is bound to the inhibitor in the presence of Me-THF and [Me-THF] is the concentration of Me-THF, in which 50% of *efmSHMT* is bound to Me-THF in the presence of the inhibitor. Here, IC₅₀ is the concentration of inhibitor added to the *efmSHMT* solution and [Me-THF] is the EC₅₀ in the presence of the inhibitor concentration. [EC₅₀ without inhibitor] is the EC₅₀ of Me-THF in the absence of the inhibitor.

Additional sentences for the competitive binding assay of the revised Results section. (Lane 344 to 361, Page 11 and 12)

Evaluation of the binding affinities of each compound against *efmSHMT*. The binding affinities of each compound against *efmSHMT* were evaluated using a binding assay that utilised the 500-nm absorption value derived from quinonoid formation in the SHMT ternary complex of SHMT, Gly and Me-THF (*efmSHMT*/Gly/Me-THF). In the presence of excess Gly, the *efmSHMT*/Gly/Me-THF complex forms quinonoid, whereas the *efmSHMT*-inhibitor complex does not. Therefore, adding a potential inhibitor to the ternary complex decreases the absorbance at 500 nm (OD₅₀₀) derived from quinonoid if the compound functions as an SHMT inhibitor (Fig. 5a). Using this feature and the Cheng-Prusoff equation, we estimated the K_i of compounds. (+)-SHIN-1 showed a strong binding affinity in the presence of 2.5 or 5 μM *efmSHMT* ($K_i = 0.0091$ and 0.033

μM , respectively) (Fig. 5b). Additionally, PMX bound to *efmSHMT* with weak affinity ($K_i = 123 \mu\text{M}$), whereas SHMT-IN-2, SER and MTX did not bind to *efmSHMT* (Fig. 5b and Supplementary Fig. S4). These results are consistent with the DSF results. However, in this assay, the detection limit concentration of *efmSHMT* was $2.5 \mu\text{M}$, and the concentration was too high to evaluate the affinity of (+)-SHIN-1 accurately. In contrast, the lowest SHMT concentration used represents the lower limit K_i of (+)-SHIN-1 determined in this assay. Therefore, the actual K_i is probably lower than $0.0091 \mu\text{M}$, indicating the extremely strong affinity of (+)-SHIN-1 toward *efmSHMT*. The DSF and the binding assay results showed that the main target of (+)-SHIN-1 is SHMT in *E. faecium*, which is the same as in humans. Furthermore, the binding affinity of (+)-SHIN-1 toward *efmSHMT* was enhanced in the presence of Ser (Supplementary Fig. S5).

Additional references

Line 827 to 829, Page 32 (reference 1 of this response letter)

Ref. 23 Stover, P. & Schirch, V. 5-Formyltetrahydrofolate polyglutamates are slow tight binding inhibitors of serine hydroxymethyltransferase. *Journal of Biological Chemistry* **266**, 1543-1550 (1991).

Line 830 to 832, Page 32 (reference 2 of this response letter)

Ref. 24 Scaletti, E., Jemth, A.S., Helleday, T. & Stenmark, P. Structural basis of inhibition of the human serine hydroxymethyltransferase SHMT 2 by antifolate drugs. *FEBS letters* **593**, 1863-1873 (2019).

17) Lane 304. Reference to Figure 3 is wrong.

Line 379, Page 12

Figure 3 was changed to Fig. 7.

Responses to Reviewer #2 comments

Major comments:

18) In humans the production of thymidine can be sustained by both SHMT1 and MTHFD1 enzymes, as they both can generate the methylene-THF substrate for TYMS. The review wonders if FOLD, the bacterial homologue of MTHFD1 can compensate for SHMT inhibition in bacteria, and encourage the authors to elaborate on this. If FOLD has relevance it should be added to Figure 2.

We appreciate the comment raised by reviewer #2. We have determined the folate binding proteins coded in the *E. faecium* genome [Genbank code: UFYJ01000001.1]. As a result, *E. faecium* utilizes SHMT as the main pathway to supply 1C units and the glycine cleavage system (GCS) as a support pathway to produce CH₂-THF(3). Interestingly, *E. faecium* has the 5,10-methylene tetrahydrofolate dehydrogenase (FolD) gene and the formyltetrahydrofolate synthetase (*Fhs*) gene. Therefore, the hypothesized folate cycle in *E. faecium* is shown in the revised Fig. 3a (see page 22 in this response letter). Furthermore, using real-time PCR (RT-PCR), we compared the mRNA amounts for folate binding proteins in the presence and absence of (+)-SHIN-1 at two time points, 4 and 10 h after adding (+)-SHIN-1. There were no drastic changes in mRNA levels for expression of folate binding protein except for thymidylate synthetase (TS) at both time points (Fig 3b in this response letter, Page 22). However, 4 h after adding (+)-SHIN-1, which was during the growth phase (Fig. 2a in this response letter, Page 21), the mRNA levels of SHMT and glycine cleavage system H protein (GcvH) increased slightly. The mRNA levels of TS, formyltetrahydrofolate synthetase (*Fhs*) and cobalamin-independent methionine synthase (*MetE*) decreased, and the mRNA levels of other folate binding proteins remained unchanged. Ten hours after adding (+)-SHIN-1, which was during the stationary phase (Fig. 2a in this response letter, Page 21), the mRNA levels of folate binding proteins, except for TS and DHFR, had increased. In particular, the mRNA amount for the expression of TS was drastically reduced. Thus, thymidine starvation occurred in *E. faecium* at this time point. The bacteriostatic rather than bactericidal effect of (+)-SHIN-1 is most likely caused by *E. faecium* recycling THF and CH₂-THF to prevent cell death, which is achieved by the cells adjusting the expression levels of these various folate proteins as mRNA quantitation experiments showed.

Previously, Alfadhli *et al.* reported (4) that *E. coli* mutants, which lacked SHMT, could not synthesize glycine and failed to proliferate in the absence of Gly (see our response to comment No. 59, Page 17 in this response letter). In this previous report, *E. coli* lacking SHMT was able to grow in the presence of excess Gly because, in *E. coli*, Gly can supply CH₂-THF through GCS and supply guanosine triphosphate (GTP) through the purine synthesis pathway. However, we observed that *E. faecium* was unable to grow in the presence of excess Gly. This difference may be derived from differences in the folate synthesis pathway between *E. coli* and *E. faecium*. Thus, *E. coli* was able to synthesise folate using GTP, but *E. faecium* lacks this pathway. Therefore, *E. faecium* was unable to synthesise folate even in the presence of excess Gly, suggesting that growth of *E. faecium* is difficult. Another possibility is that (+)-SHIN-1 inhibits both SHMT- and GCS-related proteins. Further research is required to clarify these two possibilities and is beyond the scope of this study. We have mentioned these points in the revised manuscript as follows.

Additional sentences in the paragraph on mRNA quantitation in the revised Results section.
(Line 422 to 436, Page 14 in the revised manuscript)

Quantitation of mRNA for folate binding proteins

To clarify the folate cycle of *E. faecium*, we identified folate binding proteins coded in the *E. faecium* genome (Genbank code: UFYJ01000001.1) and likely to be part of the folate cycle of *E.*

faecium (Fig. 3a). As a result, *E. faecium* uses SHMT as a primary pathway to supply 1C units and the glycine cleavage system (GCS) as a support pathway to produce CH₂-THF. Additionally, *E. faecium* was found to have a formyltetrahydrofolate synthetase (Fhs)(3), providing CH₂-THF by mediating FoD (Fig. 3a). The mRNA levels of the identified folate binding proteins in the presence and absence of (+)-SHIN-1 were compared by real-time PCR (RT-PCR) to analyse the antibacterial mechanism of (+)-SHIN-1. The mRNA levels of SHMT and glycine cleavage system H protein (GcvH) were found to increase slightly 4 h after adding (+)-SHIN-1 to *E. faecium* cells in the growth phase (Fig. 2a). GCS also supplies CH₂-THF from THF and Gly. Fhs and cobalamin-independent methionine synthase (MetE) mRNA levels decreased slightly, whereas the mRNA of other folate binding proteins did not change (Fig. 3b). The mRNA levels of folate binding proteins, except TS and DHFR, increased 10 h after adding (+)-SHIN-1 to *E. faecium* cells in the stationary phase (Fig. 2a and Fig 3b). In particular, the mRNA amount for the expression of TS decreased drastically. Thus, thymidine starvation occurred in *E. faecium* at this time point.

Additional sentences in the Discussion section of the revised manuscript (Line 463 to 473, Page 15 in the revised manuscript)

Previously, Alfadhli *et al.* reported that *E. coli* mutants that lack SHMT were unable to synthesise glycine and failed to proliferate in the absence of glycine. In a previous report, *E. coli* lacking SHMT was able to grow in the presence of excess Gly because CH₂-THF was supplied through GCS in the presence of excess Gly and also guanosine triphosphate (GTP) was provided through the purine synthesis pathway. In contrast, we showed that *E. faecium* was unable to grow in the presence of excess Gly. These observations revealed differences in the folate synthesis pathway between *E. coli* and *E. faecium*. Thus, *E. coli* synthesised folate using GTP, but *E. faecium* lacked the pathway to provide GTP. Therefore, *E. faecium* cannot synthesise folate even in the presence of excess Gly, suggesting that *E. faecium* growth in the absence of folate is hampered. Moreover, (+)-SHIN-1 may inhibit SHMT and GCS-related proteins; however, the results from the competitive binding assay indicated that the main target of (+)-SHIN-1 is *efm*SHMT. Further research is required to clarify these possibilities.

Additional sentences in the Materials and Methods section of the revised manuscript (Line 256 to 272, Page 9 in the revised manuscript)

Relative quantitation of mRNA for expression of folate binding proteins. *E. faecium* were incubated at 37°C overnight in 20 mL of MH broth with and without 2 µM of (+)-SHIN-1. Bacterial RNA was extracted using TRIzol™ Max™ Bacterial RNA Isolation Kit (Thermo Fisher Scientific, Tokyo, Japan) following the manufacturer's protocols. For the quantitative polymerase chain reaction (qPCR), the One-Step TB Green® PrimeScript PLUS RT-PCR Kit (TaKaRa, Kusatsu, Japan) was used and PCR was performed using the Thermal Cycler Dice Real Time System Lite (TaKaRa). The mRNA of D-alanine-D-alanine ligase (*ddl*) was used as a housekeeping RNA(5). The primer set for qPCR is described in Supplementary Table S1. The PCR protocol used was: reverse transcription step at 40°C for 5 min, an initial heating step at 95°C for 10 s; 40 cycles of the amplification step consisting of a 95°C denaturation period for 5 s, a 61 °C annealing (58 °C for TS, 60 °C for SHMT and 54 °C for *ddl*) and extension period for 30 s (1 min for *ddl*); and a melting phase from 65 to 95°C, following the standard protocols of the TaKaRa Dice system. The cycle threshold (C_t) was calculated by the second derivative maximum method. The relative RNA amounts of the parental strain and mutants were evaluated using the $\Delta\Delta C_t$ method and *ddl* as the housekeeping RNA. Thus, the relative transcription level (RTL) was calculated using: $RTL = 2^{-(\Delta\Delta C_t)}$

$\Delta\Delta C_t$, $\Delta\Delta C_t = \Delta C_t^{\text{SHIN-1}} - \Delta C_t^{\text{No drug}}$, where ΔC_t represents the C_t value of the RNA of each protein standardised by the C_t value of the mRNA of *ddl* (C_t^{ddl}). Thus, ΔC_t was calculated as: $\Delta C_t = C_t - C_t^{\text{ddl}}$.

Additional references in the revised manuscript

Line 860 to 862, Page 33 in the revised manuscript (reference 5 in this response letter)

Ref. 36 Dutka-Malen, S., Evers, S. & Courvalin, P. Detection of glycopeptide resistance genotypes and identification to the species level of clinically relevant enterococci by PCR. *Journal of clinical microbiology* **33**, 24-27 (1995)

Line 876 to 878, Page 34 in the revised manuscript (reference 3 in this response letter)

Ref. 42 Sah, S., Aluri, S., Rex, K. & Varshney, U. One-carbon metabolic pathway rewiring in *Escherichia coli* reveals an evolutionary advantage of 10-formyltetrahydrofolate synthetase (Fhs) in survival under hypoxia. *Journal of bacteriology* **197**, 717-726 (2015).

Line 881 to 883 Page 34 in the revised manuscript (reference 4 in this response letter)

Ref. 44 Alfadhli, S. & Rathod, P.K. Gene organization of a *Plasmodium falciparum* serine hydroxymethyltransferase and its functional expression in *Escherichia coli*. *Molecular and biochemical parasitology* **110**, 283-291 (2000).

19) *From all the data provided on sertraline it seems clear that this compound is not an SHMT inhibitor at all, and in fact the data provided to prove this in the original paper by Geeraerts et al (ref. 15) is very weak to begin with. The authors should make a stronger statement to this effect.*

As Reviewer #2 mentioned, there are no data showing that sertraline functions as an *efm*SHMT inhibitor. The additional binding assay included in the revised manuscript showed that sertraline does not bind *efm*SHMT. (See Fig 5b and Discussion, Page 24 to 27 in this response letter.)

Additional sentences in the Discussion section of the revised manuscript (Line 529 to 535, Page 17)

It was previously reported that PMX could bind to human SHMT-2 and *Arabidopsis thaliana* SHMT. Thus, PMX could work as a species-wide SHMT inhibitor. This information is also useful to develop novel more potent and specific inhibitors of the emerging anti-bacterial, -cancer and -parasite drug target. On the other hand, in this study, it was proved that PMX bound to *efm*SHMT although the binding affinity was too weak to work as an *efm*SHMT inhibitor within the concentration range exerting antibacterial activity. **Additionally, SHMT-IN-2 and SER, which are reported to be huSHMT inhibitors, never suppress SHMT activity as competitive binding assay and DSF proved.**

20) *The activity data in tables 1, 2, 3 and supplementary tables 3, 4 and 5 are expressed as mean EC_{50} s with standard deviations. While this is very common practice, it is scientifically incorrect to indicate spread of data on non-transformed EC_{50} with standard deviations, due to the logarithmic nature of the equations used to calculate the EC_{50} values. Instead, standard deviations should be used with log-transformed EC_{50} s (pEC_{50}). Alternatively data spread on non-transformed values can be indicated by confidence interval, e.g. 95% confidence interval (CI_{95}). In addition, reporting non-transformed EC_{50} s with SD could be an indication that the mean values have been calculated incorrectly also. Here the geometric mean should be used, while for log-transformed EC_{50} the 'normal' arithmetic mean should be used. Taken together, the authors should report pEC_{50} values*

with standard deviations. These are also more easy to compare.

We thank reviewer #2 for this valuable comment. As Reviewer #2 commented, pEC₅₀ values should be shown. However, to calculate the selectivity index, combination index and fold increase, EC₅₀ values are also useful. Therefore, we have used EC₅₀ and pEC₅₀ values together (See Table 1 and 2, Fig. 5b in the revised manuscript and Supplementary Table S3, S5, S6 and S7, supplementary Fig. S5.)

Minor comments:

21) Line 1: change 'potent' to 'potential'.

We have changed “Potent” to “Potential” (Line 1 in the revised manuscript).

22) Line 53: change 'supplies' to 'converts'.

We have changed the sentence. Please see our response to the comment raised by reviewer #1 (comment No. 5, Page 2) and line 52 to 54, Page 3 in the revised manuscript.

23) Line 58: change 'methanolate' to 'methotrexate'.

We have changed “methanolate” to “methotrexate” (Line 57, Page 3 in the revised manuscript).

24) Line 59: capitalize 5-fluorouracil.

We have changed “5-fluorouracil” to “5-Fluorouracil” (Line 58, Page 3 in the revised manuscript).

25) Line 60: change 'agent classes' to 'agents'.

We have changed “agent classes” to “agents” (line 59, Page 3 in the revised manuscript).

26) Line 82: if *E. faecium* can grow in folate-free medium how would SHMT inhibition have any effect, since it uses folate as substrate/cofactor?

We thank reviewer #2 for this comment. A previous report (ref. 18 of the revised manuscript) wrote that *E. faecium* “can grow in the folate-free medium supplemented with amino acids such as glycine, serine, histidine, and methionine, and purine and pyrimidine bases such as adenine, guanine and thymine, which are final products after folate and its derivatives function as C1 donors.” Under such conditions, *E. faecium* can incorporate essential molecules for survival without synthesis and folate derivatives. However, our experimental conditions differed from those published conditions. Therefore, we have deleted the sentence to avoid confusing readers.

27) Line 110: 'Hep2 was a kind gift'.

We have changed “Hep2 was kind gifts” to “Hep2 was a kind gift” (Line 109, Page 5 in the revised manuscript).

28) Line 129: How were the EC₅₀ values calculated, which equation and software were used?

We thank reviewer #2 for their comment. Data shown represent mean EC₅₀ values (±1 standard error (SE)) derived from the results of 2–6 independent experiments conducted in duplicate. Thus, four to twelve data points and the Microsoft Excel functions AVERAGE and STDEV/SQRT were used to calculate EC₅₀ and SE values, respectively. We have described these points in the Materials and Methods section of the revised manuscript as follows.

Additional sentences in the Materials and Methods section of the revised manuscript (Line 126 to 127, Page 5)

the EC₅₀ values of each test compound and standard error were calculated by Microsoft Excel function, AVERAGE and STDEV/SQRT.

29) *Line 135: remove space after Aldrich.*

We have removed the space after “Aldrich” (line 143 in the revised manuscript).

30) *Line 177: remove dash between histidine and tags.*

We have removed the dash between histidine and tags (line 185 in the revised manuscript).

31) *Line 182: at which temperature were the protein aliquots stored?*

We have added text describing the storage temperature as follows.

Lines 188–190 in the revised manuscript

BEFORE: Each protein preparation was concentrated in 20 mM HEPES buffer (pH 7.5) containing 50 mM NaCl (final concentration, 15 mg/ml).

AFTER: Each protein preparation was concentrated in 20 mM HEPES buffer (pH 7.5) containing 50 mM NaCl (final concentration, 15 to 32 mg/ml) **and stored at 4°C.**

32) *Line 201: remove dash between hanging and drop.*

We have removed the dash between “hanging” and “drop” (line 227 in the revised manuscript).

33) *Lines 220-225: this paragraph seems to be a duplication form the paragraph above.*

We have removed the paragraph on lines 220 to 225 of the previous version of the manuscript.

34) *Line 228: change 'numbers' to 'number'.*

We added new crystal structures. Therefore, “numbers” are correct.

35) *Line 242: remove dash between 100 and uM.*

We have removed the dash between 100 and μ M (line 280 in the revised manuscript).

36) *Line 253: change 14000 to 14,000.*

We have changed “14000” to “14,000” (line 300 in the revised manuscript).

37) *Line 259: remove 'those'.*

We have removed “those” on line 307 in the revised manuscript.

38) *Line 268: change 'conflict with' to 'counteract'.*

The sentence was deleted. Please see our response for comment No.12, Page 4 in this response letter.

Deleted sentence

As shown in Fig. 2b, an excess amount of Ser, which is an SHMT substrate, can conflict with (+)-SHIN-1.

39) Lines 281-290: the authors should comment on the biphasic nature of the melting curve induced by (+)-SHIN1, as this complicated calculating the melting point. This could be due to monomer/dimer melting.

We thank reviewer #2 for their comment. We have mentioned the biphasic nature of the melting curve in the Results section of the revised manuscript as follows.

The additional sentence in the Results section of the revised manuscript. (Line 340 to 342, Page 11)

The biphasic nature of the melting curve induced by (+)-SHIN-1 complicated calculating the melting point. This could be due to monomer/dimer melting.

40) Line 285: change 'never thermally stabilized' to 'did not significantly stabilize'.

We have changed “never thermally stabilized *efmSHMT*” to “did not stabilise *efmSHMT* significantly” (line 334 in the revised manuscript).

41) Line 287: change 'might bind' to 'binds'.

We have changed “might bind” to “binds” (line 337 in the revised manuscript).

42) Line 293: add 'in complex' before 'with' pyridoxal'.

We have added “in complex” before “with pyridoxal” (line 364 in the revised manuscript).

43) Line 302: remove dash between binding and site.

We have removed the dash between binding and site (line 376 in the revised manuscript).

44) Line 306: change 'looped' to 'loop'.

We have changed “looped” to “loop” (line 381 in the revised manuscript).

45) Line 379: change 'means' to 'suggests'.

We have changed “means” to “suggests” (line 495 in the revised manuscript).

46) Line 385: change 'possibly' to 'clearly'.

We have changed “possibly” to “clearly” (line 514 in the revised manuscript).

47) Line 386: remove 'substrate'.

We have removed “substrate” (line 515 in the revised manuscript).

48) Line 387: change 'side-chain' to 'side-chains'.

We have changed “side-chain” to “side chains” (line 515 in the revised manuscript).

49) Lines 394-397: PMX also binds to human SHMT2 (see PubMed ID 31127856), which should be mentioned to strengthen the observation of PMX being a species-wide SHMT inhibitor.

We thank reviewer #2 for this information. We have added the report (PubMed ID 31127856) as an additional reference in the revised manuscript. We show that the estimated binding affinity of PMX against *efmSHMT* is weak. Therefore, PMX is not an *efmSHMT* inhibitor within the concentration range exerting antibacterial activity. Taking into consideration these points, we have mentioned that PMX is a species-wide SHMT inhibitor in the Discussion section of the revised manuscript as follows.

Rewritten sentences in the Discussion section of the revised manuscript (Line 529 to 534, Page 17)

BEFORE: Two findings support the possibility of PMX working as a weak SHMT inhibitor in *E. faecium*. First, PMX displayed similar characteristics as (+)-SHIN-1 in the competition assays and slightly enforced the thermal stability of *efm*SHMT. Second, it has been reported that PMX binds weakly to Arabidopsis thaliana SHMT.

AFTER: It was previously reported that PMX could bind to human SHMT-2 and Arabidopsis thaliana SHMT. Thus, PMX could work as a species-wide SHMT inhibitor. This information is also useful to develop novel more potent and specific inhibitors of the emerging anti-bacterial, -cancer and -parasite drug target. On the other hand, in this study, it was proved that PMX bound to *efm*SHMT although the binding affinity was too weak to work as an *efm*SHMT inhibitor within the concentration range exerting antibacterial activity.

Additional reference in the References section of the revised manuscript

Line 831 to 833, Page 23 (reference 2 in this response letter)

Ref. 24 Scaletti, E., Jemth, A.S., Helleday, T. & Stenmark, P. Structural basis of inhibition of the human serine hydroxymethyltransferase SHMT 2 by antifolate drugs. *FEBS letters* **593**, 1863-1873 (2019).

50) Line 584: (C) should be (D).

We have changed “(C)” to “**d**” (line 729, Fig. 6 legend, in the revised manuscript)

51) Line 595: change 'was' to 'are'. Which isoform of huSHMT is shown, 1 or 2?

*hu*SHMT should have been *hu*SHMT2. We have changed *hu*SHMT to *hu*SHMT2. (line 742, Fig. 6 legend, in the revised manuscript) We also changed “was” to “are”. (line 742, Fig. 6 legend, in the revised manuscript)

52) Line 598: change 'showed' to 'shown'.

We have changed “showed” to “shown”. (line 744, Fig. 6 legend, in the revised manuscript)

53) Line 846: remove after (+)-SHIN1.

We have removed “after (+)-SHIN-1” (Please see supplementary Table S7).

Responses to Reviewer #3 comments

Issues of concerns

54) *In bacterial inhibition assays, how can the authors be certain that those IC metabolism inhibitors target specifically only SHMT? These inhibitors are antifolate agents which could also interfere with other folate enzymes. The molecular mechanism of bacterial inhibition is still unclear.*

We appreciate the important comment raised by reviewer #3. Because (+)-SHIN-1 is located at the folate binding site of *efmSHMT* in the crystal structure (Fig. 6 of the revised manuscript and page 24 of this response letter), it is necessary to consider the possibility that (+)-SHIN-1 functions as a promiscuous inhibitor against other folate binding proteins in *E. faecium*. Nonetheless, it is challenging to comprehensively determine the interaction between (+)-SHIN-1 and all folate binding proteins. However, an additional experiment carried out showed that (+)-SHIN-1 binds *efmSHMT* with a K_i lower than 9.1×10^{-9} M, indicating a strong interaction between (+)-SHIN-1 and *efmSHMT* (see our response to comment No. 16 by reviewer #1; page 5 to 7 of this response letter). Furthermore, in this research, we compared two compounds with very similar structures, (+)-SHIN-1 and SHMT-IN-2 (see Fig. 1 of the revised manuscript). These two compounds have been reported to inhibit human SHMT. Our results showed that SHMT-IN-2 did not bind to *efmSHMT*. Moreover, no folate binding protein recognized SHMT-IN-2, as shown in the competitive folate assay, suggesting that the chemical structures of the SHIN series cannot bind promiscuously to folate binding proteins and (+)-SHIN-1 selectively binds to *efmSHMT*. These data indicated that the main target of (+)-SHIN-1 is SHMT in *E. faecium*, as observed in humans. Additionally, we determined the time-dependent manner of bacterial viability (see our response to Comment No. 55, page 15 and 16 of this response letter). The results showed that (+)-SHIN-1 acted bacteriostatically and not as a bactericide. The bacteriostatic activity of (+)-SHIN-1 was probably because of the supporting CH₂-THF synthesis pathway mediating Fhs and FOLD enzymes or the Gly cleavage system (GCS) (see our response to Comment No. 18, page 8 to 10 and Fig. 3a in this response letter (page 22) or the revised manuscript). In the presence of (+)-SHIN-1, *E. faecium* can supply CH₂-THF through GCS and Fhs-FOLD pathways. However, the amount of CH₂-THF supplied through these support pathways was likely insufficient to maintain cell growth but sufficient to avoid cell death in the presence of (+)-SHIN-1. These molecular mechanisms have been described in the Discussion section of the revised manuscript.

NOTE: The Discussion section of the revised manuscript has been changed significantly. The Discussion section before and after revisions is presented on pages 25–27 of this response letter.

55) *I am curious about the sensitivity and accuracy of spectrophotometric measurement to quantitate the bacterial cell growth and to evaluate the EC₅₀. Measurement of turbidity is not very sensitive and could be limited at high absorptivity. The assays should also be done by measuring colony forming unit (CFU) as well.*

We have evaluated bacteria viability using Microbial Viability Assay Kit-WST (DOJINDO, Tokyo, Japan) to quantitate bacterial cell growth accurately and determine the inhibition mechanisms of the compounds (see Fig. 2 in the revised manuscript and page 21 in this response letter). The results showed that the antibacterial activity of (+)-SHIN-1 ($EC_{50} = 0.21 \pm 0.13$ nM), which was detected by measuring bacterial viability, was similar to that determined by measuring turbidity (0.044 ± 0.028 nM). Other compounds also inhibited bacterial growth over a similar concentration range to the results obtained from the turbidity experiments. Notably, the detected

time-dependent manner of bacterial viability indicated that (+)-SHIN-1 acts bacteriostatically rather than as a bactericide. This observation is because *E. faecium* has supporting (not main) CH₂-THF synthesis pathways mediating Fhs and FOLD enzymes (see our response to the comment raised by reviewer #2 (No. 18, page 8 to 10)). We have added sentences in the Material and Methods and Results sections of the revised manuscript as follows.

Additional sentences in the Materials and Methods section of the revised manuscript (Line 129 to 138, Page 5 and 6)

Determination of bacterial viability. Determination of bacterial viability. Living *E. faecium* were determined using the Microbial Viability Assay Kit-WST (DOJINDO, Tokyo, Japan). Briefly, *E. faecium* were grown overnight at 37°C on Nissui Plate Sheep Blood Agar. *E. faecium* were diluted to a turbidity equivalent that matched 1.0 McF standard in distilled water, followed by a further dilution of 1:200 in MH broth with or without drugs. Positive control wells included bacteria seeded into MH broth without compounds, and the negative control wells contained MH broth only. All experimental and control wells had a final volume of 200 µL. Five microlitres of the colouring reagent was added to each well of the 96-well plate every 2 h after starting incubation with each compound. The absorbance at 450 nm (OD₄₅₀) was measured 1 h after adding the colouring reagent using the GloMax microtiter plate reader (Promega, Tokyo, Japan). EC₅₀ values were calculated using the values of relative bacterial viability after 14 h incubation.

Additional sentences in the Results section of the revised manuscript. (Line 285 to 292, Page 10)

Determination of time dependent manner of bacterial viability with or without inhibitors. T We evaluated the time-dependent change in bacterial viability in the presence and absence of each inhibitor to quantitate the bacterial cell growth carefully and determine the inhibition mechanism of the compounds (Fig. 2) The results revealed that (+)-SHIN-1, MTX, TMP and SER inhibited the growth of *E. faecium* with similar EC₅₀ values to those of turbidity detection (EC₅₀ = 0.00023, 0.0084, 0.0088 and 13 µM, respectively). SHMT-IN-2 showed weaker activity than the turbidity detection result. Notably, time-dependent bacterial viabilities indicated that (+)-SHIN-1 acts bacteriostatically rather than function in a bactericidal manner against *E. faecium*.

56) *The manuscript lacks comprehensive studies of steady-state and inhibition kinetics of the recombinant enzyme. A lot of the data presented are indirect.*

We appreciate the important comment raised by reviewer #3. To evaluate direct binding of compounds to *efm*SHMT, we have constructed a binding assay system that detects the absorbance (500 nm) of quinonoid, which forms in the ternary complex of SHMT, Gly and 5-methyltetrahydrofolate (Me-THF) (1). Please see our response to a similar comment raised by reviewer #1 (comment No. 16, page 5 to 7 in this response letter).

Additional reference in the References section of the revised manuscript

Line 827 to 829, Page 32 (reference 1 in this response letter)

Ref. 23 Stover, P. & Schirch, V. 5-Formyltetrahydrofolate polyglutamates are slow tight binding inhibitors of serine hydroxymethyltransferase. *Journal of Biological Chemistry* **266**, 1543-1550 (1991).

57) ITC experiments can be used to evaluate direct interactions between the enzyme and ligands instead of using indirect delta Tm values.

We thank reviewer #3 for this advice. Interactions between *efm*SHMT and ligands were evaluated by constructing a binding assay system (please see our response to comments No. 16 and 56, pages 5 to 7 and 16).

58) In competition assays, concentrations of competing substances used in the assays are rather high. Are these relevant to the physiology concentrations?

We have used a competition assay to determine the change in antibacterial activity in the presence of excess concentrations of biomolecules. There are no relevant physiological concentrations available. To clarify this point, we have added text in the Result section of the revised manuscript as follows.

Rewritten sentences in the Results section of the revised manuscript (Line 303 to 306, Page 10)

BEFORE:

Competition by serine, glycine and physiological nucleosides

To identify the protein target of (+)-SHIN-1 and other compounds, we performed competition assays using folic acid, Ser, Gly, inosine and dT.

AFTER:

Antibacterial assay with excess concentration of serine, glycine and physiological nucleosides

The antibacterial mechanism of (+)-SHIN-1 and other compounds was determined by performing antibacterial assays with excess concentrations of folic acid, Ser, Gly, inosine and dT when compared with the physiological concentrations of these molecules (Fig. 3a).

59) The authors should validate that *efm*SHMT is essential for *efm* cell growth. The deletion and functional complementation analysis should be investigated or discussed.

We thank reviewer #3 for their comment. Alfadhli *et al.* reported previously that *E. coli* mutants that lacked SHMT could not synthesize glycine and failed to proliferate in the absence of Gly (4). Thus, SHMT is a critical enzyme for bacterial growth. In this previous report, *E. coli* lacking SHMT was able to grow in the presence of excess Gly because, in *E. coli*, Gly can supply CH₂-THF through the glycine cleavage system (GCS) and also supply guanosine triphosphate (GTP) through the purine synthesis pathway (3). GTP is one of the building blocks for folate synthesis. Thus, in the presence of excess Gly, *E. coli* can synthesize CH₂-THF, folate, purines and pyrimidines without active SHMT. In contrast, *E. faecium* lacks the folate synthesis system. Therefore, *E. faecium* was unable to synthesise folate in the presence of excess Gly, indicating that *E. faecium* cannot grow under these conditions. However, *E. faecium* does have GCS and Fhs-Fold pathways, which support CH₂-THF synthesis (see our response to comment No. 18, page 8 to 10 in this response letter). In the presence of (+)-SHIN-1, *E. faecium* can synthesise CH₂-THF through the GCS and Fhs-Fold pathways even though the amount of CH₂-THF produced would be insufficient to maintain cell growth but sufficient to avoid cell death, which is caused by inhibition of one-carbon metabolism. This observation is supported by the bacterial viability assay results that showed that (+)-SHIN-1 acts bacteriostatically and not as a bactericide. We have included text on this point in the Discussion section of the revised manuscript (see our response to comment No. 18, page 8 to 10 in this response letter).

60) *In addition to cell cytotoxicity, the authors should present in vivo pharmacokinetics and pharmacodynamics of (+)-SHIN-1 to evaluate dosage and safety used of (+)-SHIN-1.*

We thank reviewer #3 for this valuable comment. A previous report (6) showed that SHIN-1 lacks pharmacokinetic properties suitable for an *in vivo* study of SHMT biology. Thus, the half-life of SHIN-1 is too short to use *in vivo*. The short half-life of SHIN-1 was overcome by chemical modification. The modified SHIN-1 increased the survival rate of a mouse model. We have described this point in the Discussion section of the revised manuscript.

Additional sentences in the Discussion section of the revised manuscript. (Line 482 to 485, Page 15 and 16)

Furthermore, a previous report showed that SHIN-1 lacked pharmacokinetic properties suitable for *in vivo* study of SHMT biology, but SHIN-1 derivative, which acted as a human SHMT inhibitor, overcame that disadvantage and exerted anticancer activity *in vivo*.

Additional reference in the revised manuscript

Line 879 to 880, Page 34 (reference 6 of this response letter)

Ref. 43 García-Cañaveras, J.C., *et al.* SHMT inhibition is effective and synergizes with methotrexate in T-cell acute lymphoblastic leukemia. *Leukemia* **35**, 377-388 (2021).

61) *Although (+)-SHIN-1 shows promising results as a potent inhibitor, further modification of (+)-SHIN-1 should result in more potent inhibitors. This aspect should be discussed.*

We thank reviewer #3 for their comment. (+)-SHIN-1 and SHMT-IN-2, which were reported as huSHMT inhibitors, have similar chemical structures, but the binding characteristics against *efmSHMT* were quite different. The crystal structure of the *efmSHMT*/(+)-SHIN-1 complex suggested that a hydrogen bond between the 343-loop and exocyclic hydroxy group in (+)-SHIN-1 may be responsible for the difference in binding affinity. Additionally, Ser binding to *efmSHMT* enhanced the binding affinity of (+)-SHIN-1 by stabilizing the 115-loop. However, there were no direct interactions between (+)-SHIN-1 and Ser bound to *efmSHMT*. Thus, for the bi-phenyl moiety in (+)-SHIN-1, modifications that maintain the interaction with the 343-loop may be effective in developing novel SHMT inhibitors, which have species specificity. The modification using SHMT-IN-2 as a basic structure should facilitate the generation of huSHMT specific anticancer drugs, which have not been reported to affect intestinal flora. Furthermore, for the pyrazolopyran moiety, the enhanced affinity caused by interaction with Ser directly bound to *efmSHMT* should be useful in developing more potent SHMT inhibitors than (+)-SHIN-1. We have described these points in the Discussion section of the revised manuscript as follows.

Additional sentences in the Discussion section of the revised manuscript. (Line 520 to 528, Page 16 and 17)

Furthermore, Ser binding *efmSHMT* enhanced the binding affinity of (+)-SHIN-1 by stabilizing the 115-loop. However, there were no direct interaction between (+)-SHIN-1 and Ser binding to *efmSHMT*. Thus, for bi-phenyl moiety in (+)-SHIN-1, the modification which maintaining the interaction with 343-loop would be effective to develop novel SHMT inhibitors, which had species specificity. The modification using SHMT-IN-2 as a basic structure would enable to create huSHMT specific anticancer drugs, which never affected to intestinal flora. Furthermore, for pyrazolopyran moiety, the improvement which enable the moiety to directly interact with Ser binding to *efmSHMT* was useful to develop the more potent SHMT inhibitors than (+)-SHIN-1.

The deleted sentence in the Discussion section of the Revised manuscript.

Thus, the structural diversity in the two loops is useful for designing species specificity in the compounds, as is the case with (+)-SHIN-1 and SHMT-IN-2.

62) *The data would be clearer if the enzyme-inhibitor complex can be co-crystallized in the presence of substrate/product i.e. L-ser, Gly, THF, MTHF. Without having these complexes, the authors can compare the current structure with previously solved co-complex structures with substrates. This way, discussion on mode of inhibition can be more insightful.*

We thank reviewer #3 for the suggestion. We have added the co-complex structures of *efm*SHMT with L-Ser and (+)-SHIN-1 and with Gly and Me-THF (Fig. 6e and 6f in the revised manuscript). We describe these data in the Results and Discussion sections of the revised manuscript as follows.

Additional sentences in the Results section of the revised manuscript. (Line 363 to 367, Page 12)

We solved the structures of *efm*SHMT in complex with pyridoxal 5'-phosphate (PLP) and (+)-SHIN-1 in the absence or presence of Ser (Fig. 6, Supplementary Table S3) at 2.28 and 1.90 Å resolutions, respectively, to characterise SHIN-1 binding to *efm*SHMT. In addition to these structures, the *efm*SHMT/Gly/Me-THF complex structure was determined at 2.62 Å resolution.

Additional sentences in the Results section of the revised manuscript. (Line 412 to 420, Page 13 and 14)

In the structure of the *efm*SHMT/Ser/(+)-SHIN-1 complex, the covalent bond formed by a Schiff's base relocated from K226 to the amino group of Ser (Fig. 6e). The Ser amino acid forms hydrogen bonds with E53' (prime denotes the neighbour molecule) and H122 to stabilize the 115-loop, and the carbonyl group of Ser also forms a salt bridge with R359. Similarly, the structure of *efm*SHMT/Gly/Me-THF showed the Schiff's base between PLP and Gly (Fig. 6f). As mentioned above, *ec*SHMT and *efm*SHMT share structural similarity. The binding mode of folate (Me-THF in *efm*SHMT and formyl-THF in *ec*SHMT) is also conserved. In the structure of *efm*SHMT/Gly/Me-THF, Me-THF forms hydrogen bonds with the side chains of E53' and N343 and main chain amide groups of L117, G121 and L123 (Fig. 6f).

Additional sentences in the Discussion section of the revised manuscript. (Line 498 to 510, Page 16)

Furthermore, the competitive binding assay showed that the binding affinity of (+)-SHIN-1 to *efm*SHMT was enhanced under coexistence with Ser when compared with Gly, while that of Me-THF did not change. In the structure of the *efm*SHMT/Ser/(+)-SHIN-1 ternary complex, the serine residue on PLP formed hydrogen bonds with E53 and H122. H122 is located on the 115-loop, and this hydrogen bond stabilises the loop. Structural analysis revealed that (+)-SHIN-1 forms several hydrogen bonds with amino acids in the 115-loop. Loop stabilization contributes to the binding affinity of (+)-SHIN-1 to *efm*SHMT. In contrast, the binding affinity of Me-THF was not altered in the assay, as mentioned above. Me-THF also hydrogen bonds with amino acids in the 115-loop. Loop stabilization can also contribute to the binding affinity of Me-THF. In the *efm*SHMT/Gly/Me-THF structure, E53 formed a hydrogen bond with Me-THF. Although the loss of the hydrogen bond with the 115-loop may negatively affect the binding affinity of Me-THF, the hydrogen bond with N10 positively contributes to the binding affinity. For Me-THF binding, each contribution was countered by hydrogen bond switching, and no change in the binding affinity was observed.

63) *The authors presented the synergistic effects of (+)-SHIN-1 and other antifolates i.e. 5-FdU and MTX. I am curious how these inhibitors interact with the target enzyme.*

We appreciate the comment raised by reviewer #3. We are also interested in the interactions between these compounds and the target enzyme. However, in this research, we focused on SHMT. Therefore, the analysis of 5-FdU, MTX and target enzymes would unnecessarily reduce the focus of this manuscript and potentially confuse readers. Moreover, such a study is outside the scope of the current study. Therefore, we did not determine and discuss the interactions between these compounds and target enzymes.

64) *Besides X-ray structures presented, the authors should perform MD simulations to investigate the motion of (+)-SHIN-1 binding compared to other pyrazolopyran derivatives in other SHMT complexes.*

We thank reviewer #3 for their comment. The purpose of this study was to characterise the potential of SHMT as an antibacterial target and not to find more potent SHMT inhibitors than (+)-SHIN-1. Therefore, we did not compare (+)-SHIN-1 with other compounds using MD simulations, which we believe is outside the scope of this study. We plan to screen for SHMT inhibitors using the competitive binding assay (see our response to comment No. 16, Page 5 to 7 in this response letter) and the Tohoku University original chemical library.

65) *The authors should perform the quantitative flux analysis of (+)-SHIN-1 to ensure that this compound does directly target to only SHMT.*

We appreciate the kind advice provided by reviewer #3. In Japan, the use of isotopes is under strict control regulations. We would require a special room to work with isotopes, and analysis tools must be located in this special room. Therefore, it is currently difficult to readily perform quantitative flux analysis. We used a competitive binding assay to determine the target enzyme of (+)-SHIN-1 (see our responses to comments No. 16 and 54, page 4 and 16 in this response letter). The results revealed that (+)-SHIN-1 showed a high binding affinity toward *efm*SHMT. Furthermore, the competitive folate assay showed that SHMT-IN-2, which has a very similar chemical structure, did not interact with any folate binding proteins. This result suggests that the chemical structures of the SHIN series cannot bind promiscuously to folate binding proteins, and (+)-SHIN-1 selectively binds to *efm*SHMT. These data indicated that the main target of (+)-SHIN-1 was SHMT in *E. faecium*, as observed for SHMT from humans.

66) *Anything known about the mRNA transcription of *efm*SHMT? Could (+)-SHIN-1 affect the *efm*SHMT gene expression level?*

We thank reviewer #3 for their comment. We have determined the mRNA levels of folate binding proteins in the presence and absence of (+)-SHIN-1. Four and ten hours after adding (+)-SHIN-1, the mRNA level of *efm*SHMT increased when compared with results in the absence of (+)-SHIN-1. However, the relative ratio of mRNA levels with and without (+)-SHIN-1 was low (see Fig 3b in the revised manuscript).

Fig. 2 The results of the bacterial viability assay. **a** Bacterial viability in the absence and presence of different (+)-SHIN-1 concentrations. **b–e** Bacterial viabilities in the absence and presence of different MTX, TMP, SER and SHMT-IN-2 concentrations, respectively. These experiments were performed six times.

Supplementary Fig. S1 Effect of glycine on *E. faecium* growth. **a**) OD₄₅₀ values were shown. A hundred molar of Ser suppressed the growth of *E. faecium*. **b**) Relative bacterial viabilities were shown. EC₅₀ value at 16 hours later was between 2 and 20 nM. These experiments were performed six times.

Fig. 3 The postulated folate cycle in *E. faecium* and changes in mRNA levels of folate binding proteins upon addition of (+)-SHIN-1. **a** *E. faecium* genome [Genbank code: UFYJ01000001.1] codes for several proteins related to the folate cycle. GCS and Fhs may support production of THF and CH₂-THF under SHMT-inhibition conditions. **b** The relative mRNA expression levels of each folate binding protein with and without (+)-SHIN-1. Four hours after adding 2 μM (+)-SHIN-1, mRNAs for expression of SHMT and GcvH increased and those of Fhs, MetE and TS decreased. In contrast, 10 h after adding 2 μM (+)-SHIN-1, mRNA expression levels of folate binding proteins, except TS and DHFR, tended to increase. Open circles are the values for calculating the average values and standard error. TS: thymidylate synthetase; DHFR: dihydrofolate reductase; GCS: glycine cleavage system; MetE: cobalamin-independent methionine synthase; Fhs: formyltetrahydrofolate synthetase; Folds: 5,10-methylene tetrahydrofolate dehydrogenase; PurN: glycinamide-RNase-transformylase N; GAR: glycinamide ribonucleotide; FGAR: formyl glycinamide ribonucleotide; PurH: AICAR transformylase; AICAR: 5-amino-4-imidazole carboxamide ribotide; FAICAR: 5-formamidoimidazole-4-carboxamide ribonucleotide; GcvH: glycine cleavage system H protein.

Fig. 5 Competitive binding assay results. **a** OD₅₀₀ values with each concentration of Me-THF and 5 µM of *efm*SHMT in the presence and absence of each compound. Addition of (+)-SHIN-1 and PMX suppressed Me-THF binding to *efm*SHMT. **b** K_i of each compound and other values, which were used for calculating K_i, are shown. Stronger K_i were observed at the lower *efm*SHMT concentrations used. All experiments were performed in the presence of 10 mM Gly and 100 µM Ser. These experiments were performed three to six to ten times. pEC₅₀, pConc and pK_i are log-transformed EC₅₀, Conc. and K_i, respectively.

Supplementary Fig. S4 OD₅₀₀ values with each concentration of (+)-SHMT and 2.5 µM *efm*SHMT. These experiments were performed six times.

Supplementary Fig. S5 Competitive binding assay results. **a** Relative OD₅₀₀ at each concentration of Me-THF and 5 µM *efm*SHMT in the presence and absence of (+)-SHIN-1. All experiments were carried out under conditions with 10 mM Gly but without Ser. The binding affinity of (+)-SHIN-1 to *efm*SHMT was weaker than that observed in the presence of Ser. These experiments were performed four times. pEC₅₀, pConc and pK_i are log-transformed EC₅₀, Conc. and K_i, respectively.

Fig. 6 Crystal structure of the *efmSHMT*/(+)-SHIN-1 complex. **a** Cartoon representation of the *efmSHMT* dimer. Green (chain A) and blue (chain B) cartoon structures are *efmSHMT* monomers. Yellow-coloured stick representation structures are PLP, whereas cyan-coloured stick representation structures are (+)-SHIN-1. **b** Electron density map of (+)-SHIN-1. The $F_o - F_c$ map of (+)-SHIN-1 is contoured at 3.0 σ . **c** Polar interaction between *efmSHMT* and (+)-SHIN-1. Magenta dotted lines are hydrogen bonds or polar interactions. Exocyclic amine and hydroxy groups in (+)-SHIN-1 form hydrogen bonds with the mainchain of G121, L117 and P352. The exocyclic amine group also forms water (W1)-mediated hydrogen bonds with S344, L117, and S118. The pyrazole moiety interacts with the H122 side chain via hydrogen bonding. The exocyclic cyano group also forms polar interactions with the S344 main chain. **d** Non-polar interactions between *efmSHMT* and (+)-SHIN-1. The biphenyl moiety in (+)-SHIN-1 interacts hydrophobically with L117, L123, V129, P352 and F353 in chain A. The biphenyl moiety also interacts with Y60' and Y61' in chain B. **e** Ser bound to PLP hydrogen bonds with H122 and R359 in chain A, and E53' in chain B. **f** Polar interactions between *efmSHMT* and Me-THF. Me-THF interacts directly with L117, G121, L123 and N342 in chain A and E53' in chain B. The exocyclic amine group also forms water (W2)-mediated hydrogen bonds with S344, L117 and S118.

Discussion

Discussion

In this report, we determined that (+)-SHIN-1 functions as an SHMT inhibitor against *E. faecium* with extremely high potency and low cell cytotoxicity. (+)-SHIN-1 also exhibited synergism with 5-FdU and 5-FUrd. As previously reported, we found that 5-FdU blocks thymidine synthesis by competitively inhibiting deoxyuridine phosphatase and TS. 5-FUrd inhibited RNA synthesis as a uridine-kinase competitor. (+)-SHIN-1 inhibited SHMT and blocked the production of CH₂-THF, which is an essential biomolecule for thymidine and purine synthesis. Thus, in combination, (+)-SHIN-1 and these nucleoside analogues downregulated several pathways related to pyrimidine and purine synthesis, thereby synergistically enhancing thymidine starvation and creating a nucleotide imbalance. Additionally, a single dose of (+)-SHIN-1 acted bacteriostatically and not as a bactericide. *E. faecium* utilizes the supportive GCS and Fhs-FoD pathways to produce THF and CH₂-THF (Fig. 3a). The data suggested that (+)-SHIN-1 exerts a bacteriostatic rather than a bactericidal effect because *E. faecium* can recycle THF and CH₂-THF and prevent cell death by regulating the expression levels of folate-binding proteins, including GcvH and Fhs, as mRNA quantitation experiments showed.

Previously, Alfadhli *et al.* reported that *E. coli* mutants that lack SHMT were unable to synthesise glycine and failed to proliferate in the absence of glycine. In a previous report, *E. coli* lacking SHMT was able to grow in the presence of excess Gly because CH₂-THF was supplied through GCS in the presence of excess Gly and also guanosine triphosphate (GTP) was provided through the purine synthesis pathway. In contrast, we showed that *E. faecium* was unable to grow in the presence of excess Gly. These observations revealed differences in the folate synthesis pathway between *E. coli* and *E. faecium*. Thus, *E. coli* synthesised folate using GTP, but *E. faecium* lacked the pathway to provide GTP. Therefore, *E. faecium* cannot synthesise folate even in the presence of excess Gly, suggesting that *E. faecium* growth in the absence of folate is hampered. Moreover, (+)-SHIN-1 may inhibit SHMT and GCS-related proteins; however, the results from the competitive binding assay indicated that the main target of (+)-SHIN-1 is *efm*SHMT. Further research is required to clarify these possibilities.

The potent activity of (+)-SHIN-1 was only directed against *E. faecium* among the tested bacteria. *Enterococcus spp.* and *Lactobacillus spp.* require folate-related compounds for growth, and both can incorporate them. The other microorganisms cannot incorporate folate derivatives. Therefore, (+)-SHIN-1 would not be incorporated by other bacteria, as observed with DHFR inhibitors, suggesting that further improvements in the chemical structure of (+)-SHIN-1 are required for developing novel antibiotics that exhibit potent activities against a broad range of bacteria. Nevertheless, as shown using the competitive assay, the antibacterial mechanism underlying SHMT inhibition involves thymine and, in part, purine starvation. These antibacterial mechanisms share similarities with TMP, which exerts potent activities against Gram-positive and -negative bacteria. Furthermore, a previous report showed that SHIN-1 lacked pharmacokinetic properties suitable for the *in vivo* study of SHMT biology, but a SHIN-1 derivative, which acted as a human SHMT inhibitor, overcame these pharmacokinetic issues and exerted effective anticancer activity *in vivo*. Therefore, SHMT inhibition is a potential target of novel antibiotics.

The crystal structure of the SHMT/(+)-SHIN-1 complex revealed that (+)-SHIN-1 binds strongly to *efm*SHMT by forming several hydrogen bonds and hydrophobic interactions. The pyrazolopyran core of (+)-SHIN-1 interacts with the catalytic site of *efm*SHMT by forming hydrogen bonds with the main chain. A hydrogen bond formed with the main chain of Leu123, located in the 115-loop, is highly conserved among SHMTs. These hydrogen bonds should contribute to the high genetic barrier of (+)-SHIN-1. Darunavir, a protease inhibitor used for human immunodeficiency virus (HIV) infectious diseases, also forms strong hydrogen bonds with mainchain residues in the catalytic site of the HIV

protease. These hydrogen bonds contribute to the high genetic barrier of Darunavir. Indeed, *in vitro* drug-resistance induction suggests that Darunavir prevents HIV-1 protease developing significant resistance using *in vitro* drug-resistance induction. Thus, the pyrazolopyran core of (+)-SHIN-1 should prevent *efmSHMT* developing resistance by forming hydrogen bonds with the main chain of *efmSHMT*. Furthermore, the competitive binding assay showed that the binding affinity of (+)-SHIN-1 to *efmSHMT* was enhanced under coexistence with Ser when compared with Gly, while that of Me-THF did not change. In the structure of the *efmSHMT*/(+)-SHIN-1/Ser ternary complex, the serine residue on PLP formed hydrogen bonds with E53 and H122. H122 is located on the 115-loop, and this hydrogen bond stabilises the loop. Structural analysis revealed that (+)-SHIN-1 forms several hydrogen bonds with amino acids in the 115-loop. Loop stabilization contributes to the binding affinity of (+)-SHIN-1 to *efmSHMT*. In contrast, the binding affinity of Me-THF was not altered in the assay, as mentioned above. Me-THF also hydrogen bonds with amino acids in the 115-loop. Loop stabilization can also contribute to the binding affinity of Me-THF. In the *efmSHMT*/Me-THF/Gly structure, E53 formed a hydrogen bond with Me-THF. Although the loss of the hydrogen bond with the 115-loop may negatively affect the binding affinity of Me-THF, the hydrogen bond with N10 positively contributes to the binding affinity. For Me-THF binding, each contribution was countered by hydrogen bond switching, and no change in the binding affinity was observed. Notably, we found that in *E. faecium*, (+)-SHIN-1 and SHMT-IN-2 exhibit quite distinct antibacterial mechanisms, and SHMT-IN-2 did not bind to *efmSHMT* despite both compounds having the same pyrazolopyran core. These two compounds have different substructures at the biphenyl moiety position in (+)-SHIN-1. Thus, the chemical structure at this position is clearly important for the specificity of SHMT. We found that the biphenyl moiety in (+)-SHIN-1 mainly interacts hydrophobically with the aa side chains in the 115- and 343-loops. The exocyclic hydroxy group in (+)-SHIN-1 clearly interacts with the 342-loop. As reported for *pvSHMT*, the structure of the 343-loop is highly flexible. Additionally, the 115- and 343-loops share low sequence homology. The biphenyl and hydroxy groups in (+)-SHIN-1 stabilised *efmSHMT*/(+)-SHIN-1 binding by reducing the flexibility of the 343-loop. However, SHMT-IN-2 neither formed a hydrogen bond with the 343-loop nor bound to *efmSHMT*. Furthermore, Ser binding to *efmSHMT* enhanced the binding affinity of (+)-SHIN-1 by stabilizing the 115-loop. However, there were no direct interactions between (+)-SHIN-1 and Ser complexed with *efmSHMT*. Thus, for the bi-phenyl moiety in (+)-SHIN-1, developing modifications that maintain the interaction with the 343-loop should generate novel, effective SHMT inhibitors that display species specificity. The modification using SHMT-IN-2 as a basic structure would facilitate the generation of *huSHMT* specific anticancer drugs, which do not affect intestinal flora. Furthermore, developing the pyrazolopyran moiety for direct interaction with Ser complexed with *efmSHMT* should yield more potent SHMT inhibitors than (+)-SHIN-1.

It was previously reported that PMX could bind to human SHMT-2 and *Arabidopsis thaliana* SHMT. Thus, PMX could work as a species-wide SHMT inhibitor. This information is also useful to develop novel more potent and specific inhibitors of the emerging anti-bacterial, -cancer and -parasite drug target. On the other hand, in this study, it was proved that PMX bound to *efmSHMT* although the binding affinity was too weak to work as an *efmSHMT* inhibitor within the concentration range exerting antibacterial activity. Additionally, SHMT-IN-2 and SER, which are reported to be *huSHMT* inhibitors, never suppress SHMT activity as competitive binding assay and DSF proved. SER reportedly works as an efflux inhibitor against *E. coli* and probably against *E. faecium*. SHMT-IN-2 also failed to bind to SHMT and did not compete against any competitor we tested. This observation suggests that SHMT-IN-2 will likely fail to inhibit SHMT and other enzymes involved in folate-mediated 1C metabolism. Further research is required to determine the detailed mechanism underlying

the antibacterial activity of SHMT-IN-2.

In our results, the SHMT inhibitor synergistically enhanced the antibacterial activities of nucleoside analogues. Hydroxyurea, which decreases purine triphosphate levels by inhibiting ribonucleotide reductase, reportedly enhances the type 1 anti-human immunodeficiency virus activity of azido-3'-deoxythymidine, 2',3'-dideoxycytidine and, in particular, 2',3'-dideoxyinosine. SARS-CoV-2 is reported to induce purine synthesis, supporting massive viral subgenomic RNA by hijacking serine metabolism through human SHMT1. Therefore, SHMT inhibitors might enhance the anti-SARS-CoV-2 activities of RNA-dependent RNA polymerase inhibitors such as remdesivir. Thus, SHMT inhibitors may also play important roles in combination therapies for bacterial and viral infections.

In conclusion, we have shown that SHMT is a novel antibacterial target in *E. faecium*. New antibiotic classes are urgently required to develop therapeutics that possess potent activities against multi-drug resistant bacteria and cannot induce cross-resistance to currently approved antibiotics. We found that (+)-SHIN-1 exerts strong antibacterial activity and combining (+)-SHIN-1 and nucleoside analogues produced strong synergistic effects. The crystal structure of the *efm*SHMT/(+)-SHIN-1 complex showed that the pyrazolopyran core of (+)-SHIN-1 forms several hydrogen bonds with the main chain of *efm*SHMT and that the biphenyl moiety and hydroxy group of (+)-SHIN-1 are important stabilisers of the interaction between *efm*SHMT and (+)-SHIN-1 because these moieties interact and reduce the flexibility of the 115- and 343-loops. Our findings should facilitate the development of SHMT inhibitors as therapeutic agents for bacterial, viral and parasite infections and for treating cancer.

Changes to the Abstract

BEFORE:

Serine hydroxymethyltransferase (SHMT) supplies a 1-carbon (1C) unit to 1C metabolism by producing 5,10-methylene-tetrahydrofolate (CH₂-THF) from tetrahydrofolate with serine to glycine conversion. SHMT-supplied glycine and CH₂-THF are important for pyrimidine, purine and amino acid synthesis. SHMT is a potential drug target in parasites, viruses and cancer. (+)-SHIN-1 was developed for cancer therapy and is a human SHMT inhibitor. However, the potential potency of SHMT as a target of antibacterial agents is unknown. Here, we show the effectiveness of SHMT as a novel antibacterial target using (+)-SHIN-1 and *Enterococcus faecium*. We observed a synergistic effect between the SHMT inhibitor and 1C metabolism inhibitors. (+)-SHIN-1 inhibited *E. faecium* at a 50% effective concentration of 10⁻¹¹ M. SHMT inhibition mainly induced thymidine and partial purine starvation in *E. faecium*. Thermal stability and crystal structure analysis showed that (+)-SHIN-1 tightly bound to *E. faecium* SHMT (*efm*SHMT). Additionally, two variable loops in SHMT were crucial for SHMT specifically binding to inhibitors. (+)-SHIN-1 also synergistically enhanced the antibacterial activities of several nucleoside analogues. Our findings highlight the potency of SHMT as an antibacterial target, and the possibility of developing SHMT inhibitors for treating bacterial, viral and parasitic infections, and cancer.

AFTER:

Serine hydroxymethyltransferase (SHMT) produces 5,10-methylenetetrahydrofolate (CH₂-THF) from tetrahydrofolate with serine to glycine conversion. SHMT is a potential drug target in parasites, viruses and cancer. (+)-SHIN-1 was developed as a human SHMT inhibitor for cancer therapy. However, the potential of SHMT as an antibacterial target is unknown. Here, we show that (+)-SHIN-1 bacteriostatically inhibits the growth of *Enterococcus faecium* at a 50% effective concentration of 10⁻¹¹ M and synergistically enhances the antibacterial activities of several nucleoside analogues. Our results, including crystal structure analysis, indicate that (+)-SHIN-1 binds tightly to *E. faecium* SHMT (*efm*SHMT). Two variable loops in SHMT are crucial for inhibitor binding, and serine binding to *efm*SHMT enhances the affinity of (+)-SHIN-1 by stabilising the loop structure of *efm*SHMT. The findings highlight the potency of SHMT as an antibacterial target and the possibility of developing SHMT inhibitors for treating bacterial, viral and parasitic infections and cancer. (147/150 words)

1. **Stover P, Schirch V.** 1991. 5-Formyltetrahydrofolate polyglutamates are slow tight binding inhibitors of serine hydroxymethyltransferase. *Journal of Biological Chemistry* **266**:1543-1550.
2. **Scaletti E, Jemth AS, Helleday T, Stenmark P.** 2019. Structural basis of inhibition of the human serine hydroxymethyltransferase SHMT 2 by antifolate drugs. *FEBS letters* **593**:1863-1873.
3. **Sah S, Aluri S, Rex K, Varshney U.** 2015. One-carbon metabolic pathway rewiring in *Escherichia coli* reveals an evolutionary advantage of 10-formyltetrahydrofolate synthetase (Fhs) in survival under hypoxia. *Journal of bacteriology* **197**:717-726.
4. **Alfadhli S, Rathod PK.** 2000. Gene organization of a *Plasmodium falciparum* serine hydroxymethyltransferase and its functional expression in *Escherichia coli*. *Molecular and biochemical parasitology* **110**:283-291.
5. **Dutka-Malen S, Evers S, Courvalin P.** 1995. Detection of glycopeptide resistance genotypes and identification to the species level of clinically relevant enterococci by PCR. *Journal of clinical microbiology* **33**:24-27.
6. **García-Cañaveras JC, Lancho O, Ducker GS, Ghergurovich JM, Xu X, da Silva-Diz V, Minuzzo S, Indraccolo S, Kim H, Herranz D.** 2021. SHMT inhibition is effective and synergizes with methotrexate in T-cell acute lymphoblastic leukemia. *Leukemia* **35**:377-388.

REVIEWERS' COMMENTS:

Reviewer #1 (Remarks to the Author):

I think that the authors have satisfactorily answered all my questions.
In my opinion, the manuscript can be accepted for publication.

Reviewer #2 (Remarks to the Author):

The comments and concerns raised during the first round have been largely addressed satisfactorily by the authors, with the exception of the reporting of EC50 values, which seemed to have caused more confusion.

In Tables 1 and 3, and Fig 5B please report 'EC50 (μM) (pEC50 \pm SE)', so do not report SE on the EC50 values itself, but only on the pEC50s. Also make sure that the mean EC50 values are calculated by averaging the pEC50 values and then transforming the average pEC50 back to EC50, not by averaging the EC50 values itself. The same applies to the reported Ki values in Fig 5B. Provided that these changes are implemented the reviewer recommends the manuscript for publication.

Reviewer #3 (Remarks to the Author):

The authors have addressed the issues of concern well.

May 1st, 2022

Re: COMMSBIO-21-2371-T (Serine hydroxymethyltransferase as a potent target of antibacterial agents acting synergistically with one-carbon metabolism-related inhibitors)

Dear Reviewers,

Please find the attached files of our revised manuscript entitled “Serine hydroxymethyltransferase as a potential target of antibacterial agents acting synergistically with one-carbon metabolism-related inhibitors”, co-authored by Yuko Makino *et al.* We are pleased to have received favourable reviews from you and the three reviewers. We are also thankful for the comments made by the three reviewers. We provide point-by-point responses to the comments raised by the reviewers (see below).

Responses to Reviewer #1 comments

We appreciate the comment by Reviewer #1 that our manuscript can be accepted for publication. We thank Reviewer#1 again for the helpful review and comments.

Responses to Reviewer #2 comments

We appreciate the comment by Reviewer #1 that the comments and concerns raised during the first round have been largely addressed satisfactorily.

Comment:

1) In Tables 1 and 3, and Fig 5B please report ' EC_{50} (μM) ($pEC_{50} \pm SE$)', so do not report SE on the EC_{50} values itself, but only on the pEC_{50} s. Also make sure that the mean EC_{50} values are calculated by averaging the pEC_{50} values and then transforming the average pEC_{50} back to EC_{50} , not by averaging the EC_{50} values itself. The same applies to the reported K_i values in Fig 5B. Provided that these changes are implemented the reviewer recommends the manuscript for publication.

We thank Reviewer #2 for the helpful comment. We ensured that the mean EC_{50} values were calculated by averaging the pEC_{50} values and then transforming the average pEC_{50} back to EC_{50} , not by averaging the EC_{50} values itself. Please see Table 1, 2, and supplementary Table S2, S4-S6 in the revised manuscript. We also change K_i value in Fig. 5B in the revised manuscript.

Responses to Reviewer #3 comments

We appreciate the comment by Reviewer #3 that we have addressed the issues of concern well. We thank Reviewer#3 again for the helpful review and comments.